

# A new species of *Oligodon* Fitzinger, 1826 from the Langbian Plateau, southern Vietnam, with additional information on *Oligodon annamensis* Leviton, 1953 (Squamata: Colubridae)

Hung Ngoc Nguyen[1,2], Bang Van Tran[1], Linh Hoang Nguyen[1], Thy Neang[3], Platon V. Yushchenko[4] and Nikolay A. Poyarkov[4,5]

[1] Department of Zoology, Southern Institute of Ecology, Vietnam Academy of Science and Technology, Ho Chi Minh City, Vietnam
[2] School of Life Science, National Taiwan Normal University, Taipei, Taiwan
[3] Wild Earth Allies, Phnom Penh, Cambodia
[4] Faculty of Biology, Department of Vertebrate Zoology, Moscow State University, Moscow, Russia
[5] Laboratory of Tropical Ecology, Joint Russian-Vietnamese Tropical Research and Technological Center, Hanoi, Vietnam

Corresponding authors
Hung Ngoc Nguyen,
nguyen.hung.uns@gmail.com
Nikolay A. Poyarkov,
n.poyarkov@gmail.com

## ABSTRACT

We describe a new species of *Oligodon* from the highlands of the Langbian Plateau, southern Truong Son Mountains, Vietnam, based on morphological and molecular phylogenetic analyses. The new species, *Oligodon rostralis* **sp. nov** is distinguished from its congeners by the following morphological characters: medium size in adults (male TL = 582 mm); small and broad head with long protruding snout; dorsal scale row formula 15-15-13; 167 ventrals, 47 subcaudals; single preocular, single postocular; loreal and presubocular absent; six supralabials, third and fourth entering orbit; six infralabials, anterior four contacting first pair of chin shields; internasals separate from prefrontals; nasal divided; single anterior and two posterior temporals; cloacal plate undivided; hemipenes short, bilobed, bifurcating in anterior one third of their length, extending to 8th subcaudal, lacking spines and papillae, with a prominent transverse flounces and distal calyces; six maxillary teeth, the posterior three enlarged; dorsal pattern consisting of 14+4 large dark-brown blotches and a bright-orange vertebral stripe on tail and dorsum; and ventral surfaces in life cream laterally with dark quadrangular spots; dark temporal streak present, edged with white. We also provide additional information on *O. annamensis*, including a morphological dataset of all specimens known from natural history collections and confirmation of an earlier record of *O. annamensis* from Cambodia. We also provide the first record of *O. annamensis* for Dak Lak Province. Phylogenetic analyses of mtDNA genes (3,131 bp of 12S rRNA, 16S rRNA and cyt *b*) suggest sister relationships of *Oligodon rostralis* **sp. nov.** and *O. annamensis* and place them in one clade with the *O. cyclurus* and *O. taeniatus* species groups, which is concordant with previous studies on the phylogenetic relationships of *Oligodon*. Our study demonstrates high level of herpetofaunal diversity and endemism of Langbian Plateau and further supports the importance of this area for conservation herpetofaunal diversity in Indochina.

## INTRODUCTION

Located in middle of the Southeast Asian biodiversity hotspot, the Langbian Plateau is known as a local center for herpetofaunal endemism. It is inhabited by numerous species of amphibians and reptiles, many of which were only described recently (*Duong et al., 2018*; *Nazarov et al., 2012*; *Poyarkov et al., 2014*; *Poyarkov et al., 2015a*; *Poyarkov et al., 2015b*; *Poyarkov et al., 2017*; *Poyarkov et al., 2019b*; *Stuart et al., 2011*; *Rowley et al., 2016*; *Vassilieva et al., 2014*). The Kukri Snakes of the genus *Oligodon* Fitzinger, 1826, are one of the most speciose and taxonomically problematic colubrid snake groups distributed in South and Southeast Asia, with over 79 species described (*Green, Orlov & Murphy, 2010*; *Wallach, Williams & Boundy, 2014*; *Uetz, Freed & Hošek, 2019*). Due to their secretive behavior (*Tillack & Günther, 2009*), many species are known from only a few specimens or even only the holotype. Consequently, knowledge regarding *Oligodon* taxonomy, distribution, morphological variation and natural history is limited (*Leviton, 1953*; *Leviton, 1960*; *Pauwels et al., 2002*; *David, Das & Vogel, 2008*; *Neang, Grismer & Daltry, 2012*). In Vietnam 23 species of *Oligodon* have been recorded up to date, with six of them being country endemic, while eight species were described within the last decade (*David, Das & Vogel, 2008*; *David et al., 2012*; *Nguyen et al., 2016*; *Nguyen et al., 2017*; *Vassilieva et al., 2013*; *Vassilieva, 2015*). This suggests that our knowledge of *Oligodon* diversity in the Indochinese region is still far from complete.

One of the least known and enigmatic *Oligodon* species from Indochina is *Oligodon annamensis Leviton, 1953* which was described based on a single female specimen collected from ''Blao, Haut Donai'' in the Langbian plateau (currently Bao Loc, Lam Dong Province, south Vietnam) (*Leviton, 1953*; *Leviton, 1960*). *Leviton (1953)* was puzzled by the taxonomic and phylogenetic affinities of this species, and only after examining a second male specimen he assumed that *O. annamensis* might be a part of the ''*taeniatus–cyclurus*–complex'' (*Leviton, 1960*). The only other existing record of this species was recently published by *Neang & Hun (2013)*, who reported a subadult specimen identified as *Oligodon annamensis* from Phnom Samkos Wildlife Sanctuary of the Cardamom Mountains in southwest Cambodia; over 600 km westwards from the type locality (*Neang & Hun, 2013*). However, identification of the Cambodian specimen was tentative and has not been confirmed by molecular analyses; no information on the phylogenetic position of *O. annamensis* is available until this paper.

During our recent surveys in Lam Dong and Dak Lak provinces of southern Vietnam we collected two *Oligodon* specimens superficially similar in morphology with description of *O. annamensis*. However, after a closer examination of specimens from Vietnam and Cambodia, comparison of diagnostic morphological traits and phylogenetic analyses of 3,131 bp of mtDNA, we were able to identify the Dak Lak and Cambodian specimens as *O. annamensis*, while the *Oligodon* specimen from Lam Dong Province showed a unique

combination of morphological characters that differ it significantly from all other *Oligodon* taxa. Furthermore, the phylogenetic analyses of mtDNA markers suggest that the Lam Dong *Oligodon* sp. represents a distinct phylogenetic lineage, not conspecific to any other *Oligodon* sequences available. Herein it is assigned to a new species, which is described below.

## MATERIALS AND METHODS

### Nomenclatural acts

The electronic version of this article in Portable Document Format (PDF) will represent a published work according to the International Commission on Zoological Nomenclature (ICZN), and hence the new names contained in the electronic version are effectively published under that Code from the electronic edition alone (see Articles 8.5–8.6 of the Code). This published work and the nomenclatural acts it contains have been registered in ZooBank, the online registration system for the ICZN. The ZooBank LSIDs (Life Science Identifiers) can be resolved and the associated information can be viewed through any standard web browser by appending the LSID to the prefix http://zoobank.org/. The LSID for this publication is as follows: urn:lsid:zoobank.org:pub:51B851C2-5D34-4065-86EA-CF18DDD94419. The online version of this work is archived and available from the following digital repositories: PeerJ, PubMed Central and CLOCKSS.

**Sampling.** The adult male specimen of *Oligodon* sp. was collected by Bang Van Tran and Linh Hoang Nguyen during the field trip in June 2017 in Bidoup–Nui Ba National Park (hereafter NP), Lam Dong Province, Vietnam (locality 1; Fig. 1). After euthanasia with 20% solution of benzocaine, the specimen was initially preserved in 95% alcohol for one day then subsequently stored in 70% alcohol. Additional specimens of *Oligodon annamensis* were collected in Chu Yang Sin NP, Dak Lak Province, southern Vietnam, by Nikolay A. Poyarkov (locality 3; Fig. 1); and in Phnom Samkos Wildlife Sanctuary (hereafter WS) of the Cardamom Mountains, Pursat Province, southwest Cambodia by Seiha Hun (locality 4; Fig. 1); both records made in April, 2012. Geographic position of the surveyed localities is shown in Fig. 1. Because of certain morphological similarity with *Oligodon annamensis* in coloration and scalation (*Orlov et al., 2010*), we also included sequences of *O. lacroixi* in our phylogenetic analysis based on a specimen from Phu Tho Province, northern Vietnam.

Specimen collection protocols and animal operations followed the Institutional Ethical Committee of Southern Institute of Ecology, Vietnamese Academy of Science and Technology (certificate number 114/QD-STHMN of November 8, 2016).

Field work, including collection of samples and animals in the field, was authorized the Bureau of Forestry, Ministry of Agriculture and Rural Development of Vietnam (permits Nos. 170/ TCLN–BTTN of 07/02/2013; 400/TCLN-BTTN of 26/03/2014; 831/TCLN–BTTN of 05/07/2013) and Forest Protection Department of the Peoples' Committee of Dak Lak Province (permit No. 388/SNgV-LS of 24/04/2019); the fieldwork in Bidoup–Nui Ba NP was conducted under scope of the contract between Sustainable Nature Resource Management Project (SNRM) under Japan International Cooperation Agency and Southern Institute of Ecology to perform the ''Biodiversity Baseline Survey'' project of September 24, 2018.

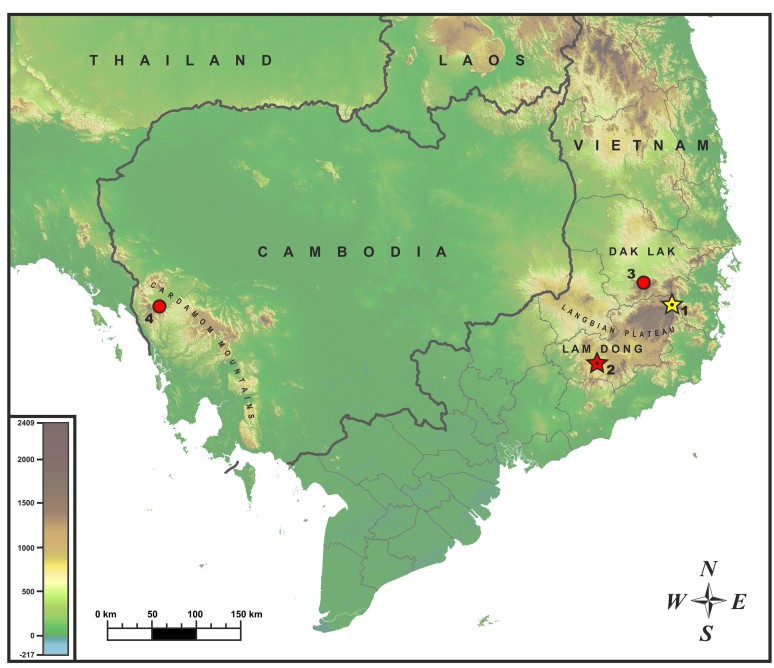

**Figure 1** **Known distribution of *Oligodon annamensis* Leviton, 1953 (red) and *Oligodon rostralis* sp. nov. (yellow) in Indochina.** Star and dot in the center of icon denotes type locality. Localities: (1) Bidoup-Nui Ba NP, Lam Dong Province, Vietnam (type locality of *Oligodon rostralis* **sp. nov.**); (2) Bao Loc (formerly "Blao, Haut Donai"), Lam Dong Province, Vietnam (type locality of *O. annamensis*); (3) Chu Yang Sin NP, Dak Lak Province, Vietnam; (4) Phnom Samkos WS, Pursat Province, Cambodia.

**Morphological analysis.** Color characters and patterns were recorded during examination of specimens in life and taken from digital images of the living specimens. Morphological characters and morphometric ratios considered to be of taxonomic importance for *Oligodon* were used for species descriptions and followed a number of recent revisions of the genus (*David, Das & Vogel, 2008*; *David et al., 2012*; *Leviton, 1953*; *Leviton, 1960*; *Neang & Hun, 2013*; *Nguyen et al., 2016*; *Nguyen et al., 2017*; *Vassilieva et al., 2013*; *Vassilieva, 2015*). All body measurements, except body and tail lengths, were taken under a binocular microscope using digital slide-caliper to the nearest 0.1 mm. Body and tail lengths were measured to the nearest millimetre with a measuring tape. The right hemipenis was forcedly everted by using water injection prior the preservation of the specimen. Methodology of ventral and subcaudal scales counts followed *Dowling (1951)*. Terminology for hemipenial structures generally followed *Smith (1943)* and *Dowling & Savage (1960)*. Maxillary teeth of the specimens were counted by examining both maxillae, directly by pushing back the soft tissue with a needle under binocular microscope prior to preservation.

The following measurements (all in mm) and counts were taken: snout to vent length (SVL)—measured from the tip of the snout to the vent; tail length (TaL)—measured from the vent to the tip of the tail; total length (TL)—sum of SVL and TaL; relative tail length to total length (RTL) calculated as tail length to total length ratio (TaL/TL); head

length (HL) from the tip of the snout to the posterior margin of the mandible; head width (HW) measured at the widest part of the head immediately posterior to the eye; head width to head length ratio (HW/HL); snout length (SnL)—distance between the tip of the snout and anterior edge of eye; eye diameter (EyeL)—maximal horizontal length of the eye; frontal scale length/width (FrL/FrW)—length and width of the frontal scale; interorbital distance (IOD)—the shortest distance between the eyes; internarial distance (IND)—distance between the nostrils; number of maxillary teeth (DEN); dorsal scale rows at neck (ASR)—number of scale rows at one head length behind the head; midbody scale rows (MSR)—number of scale rows at midbody; dorsal scale rows anterior to the vent (PSR)—number of dorsal scale rows at one head length prior to the vent; dorsal scale rows formula (DSR)—referred to as a general scale formula in the form "ASR-MSR-PSR" (for number of dorsal scale rows at neck, midbody and prior to vent, respectively); first dorsal scale reduction (RED1)—the first reduction of dorsal scale rows, corresponding to a ventral scale; ventral scales (VS)—number of scales from the second ventral scale posterior to gulars to the vent excluding cloacal plate; cloacal plate (AP)—number of terminal ventral scales immediately anterior to vent; subcaudal scales (SC)—number of paired subcaudal scales excluding the terminal scute; total belly scales (Total Sc.)—sum of ventral and subcaudal scales; supralabials (SL)—number of scales on upper lip; SL-Eye—number of SL entering orbit; infralabials (IL)—number of scales on lower lip; infralabials contacting each other (IL-contact)—number of pairs of infralabial scales in contact; infralabials contacting the anterior chin shields (IL-CS)—infralabial scales contacting the upper chin shields; number of preocular scales (PrO); number of presubocular scales (PrsO); number of postocular scales (PtO); number of anterior temporals (Ate)—temporal scales which contact the postocular scales; number of posterior temporals (Pte)—temporal scales immediately contacting the anterior temporal scales; condition of loreal scale (LOR)—1–present, 0–absent, *–vestigial; condition of nasal scale (NAS)—D–vertically divided, E–entire, PD–partially divided; hemipenis shape—(1) unforked, a single organ with no lobes at apex; (2) bilobed, organ contains two lobes at its apex; hemipenis ornamentation—notes on ornamentation of organ (i.e., spinules, calyces or flounces); presence of appendages seen *in situ* (papillae sensu *Smith, 1943*); hemipenis length—length of the everted hemipenis in mm and relative to number of subcaudal scales. Symmetric characters are given in left/right order. Other abbreviations: a.s.l.: above sea level; Div.: Division; Comm.: Commune; Dist.: District; Mt.: mountain; NP: National Park; NR: Nature Reserve; Prov.: Province; WS: Wildlife Sanctuary.

The type material was deposited in the herpetological collection of the Department of Zoology, Southern Institute of Ecology (SIEZC) in Ho Chi Minh City, Vietnam. Additional material used for comparisons is stored in the herpetological collections of Centre for Biodiversity Conservation of the Royal University of Phnom Penh, Phnom Penh, Cambodia (CBC RUPP); United States National Museum, Washington, D.C., USA (USNM); Museum National d'Histoire Naturelle, Paris, France (MNHN) and Zoological Museum of Lomonosov Moscow State University, Moscow, Russia (ZMMU).

**Molecular analyses.** Total genomic DNA was extracted from muscle tissue preserved in 95% ethanol using the Qiagen DNAeasy Blood & Tissue Kit following manufacturers'
**Table 1 Primers used in this study.**

| Gene | Primer name | Reference | Sequence |
|------|-------------|-----------|----------|
| 12S rRNA | 12S2LM | *Green, Orlov & Murphy (2010)* | 5′-ACACACCGCCCGTCACCCT-3′ |
| | 16S5H | *Green, Orlov & Murphy (2010)* | 5′-CTACCTTTGCACGGTTAGGATACCGCGGC-3′ |
| 16S rRNA | 16S1LM | *Green, Orlov & Murphy (2010)* | 5′- CCGACTGTTGACCAAAAACAT-3′ |
| | 16SH1 | *Green, Orlov & Murphy (2010)* | 5′-CTCCGGTCTGAACTCAGATCACGTAGG-3′ |
| cyt *b* | H14910 | *Dahn et al. (2018)* | 5′-GACCTGTGATMTGAAAAACCAYCGTT-3′ |
| | THRSN2 | *Dahn et al. (2018)* | 5′-CTTTGGTTTACAAGAACAATGCTTTA-3′ |

protocol. We used the polymerase chain reaction (PCR) to amplify two fragments of mitochondrial DNA (hereafter mtDNA): the first fragment including partial sequences of 12S ribosomal RNA (rRNA), tRNA-Valine and 16 rRNA genes (total length up to 2035 bp) and a complete sequence of cytochrome *b* gene (1,096 bp). Primers used for both PCR and sequencing are summarized in Table 1.

PCR protocol for 12S–16S rRNA mtDNA fragment in general followed *Green, Orlov & Murphy (2010)*. For both primer pairs of 12S and 16S rRNA, we used the following PCR protocol: (1) initial denaturation step at 94 °C for 5 min; (2) 35 cycles of denaturation at 94 °C for 1 min, annealing at 55 °C for 1 min and extension at 72 °C for 1 min; (3) final extension at 72 °C for 10 min; and (4) cooling step at 4 °C for storage.

For cytochrome *b* sequences (fragment up to 1,096 bp) we used a modified PCR protocol of *Dahn et al. (2018)* with touchdown: (1) initial denaturation step at 94 °C for 5 min; (2) 10 cycles of denaturation at 94 °C for 1 min, annealing for 1 min with temperature decreasing from 50 °C to 45 °C (with cool-down at 0.5 °C per each cycle) and extension at 72 °C for 1 min; (3) 24 cycles of denaturation at 94 °C for 1 min, annealing at 45 °C for 1 min and extension at 72 °C for 1 min; (4) final extension at 72 °C for 10 min; and (5) cooling step at 4 °C for storage.

All PCR products were sequenced in both directions by Genomics BioSci & Tech Corp. (Taipei, Taiwan). Sequences were assembled and checked using sequencher 4.9 (GeneCodes). The obtained sequences are deposited in GenBank under the accession numbers MN395601–MN395604 and MN396762; MN396765 (Table 2).

**Phylogenetic analyses.** The 12S–16S rRNA datasets of *Green, Orlov & Murphy (2010)*, *Pyron et al. (2013)*, our newly obtained sequences and other *Oligodon* sequences available in GenBank were used to examine the position of the Lam Dong *Oligodon* sp. in the matrilineal genealogy of the genus (summarized in Table 2). In total, we analysed mtDNA sequence data for 52 specimens, including 43 samples of ca. 24 species of *Oligodon*, and eight outgroup sequences of other Colubrinae representatives, and sequences of *Hebius vibakari* (Natricinae) which were used to root the tree.

Nucleotide sequences were initially aligned in MAFFT v.6 (*Katoh et al., 2002*) with default parameters, and subsequently checked by eye in BioEdit 7.0.5.2 (*Hall, 1999*) and slightly adjusted. MODELTEST v.3.6 (*Posada & Crandall, 1998*) was applied to estimate optimal evolutionary models for the data set analysis. Mean uncorrected genetic distances ($p$-distances) were calculated in MEGA 6.0 (*Tamura et al., 2013*).

Nguyen et al. (2020), *PeerJ*, DOI 10.7717/peerj.8332

**Table 2** Sequences and voucher specimens of Oligodon and outgroup taxa used in this study.

| No. | Sample ID | Genbank AN | Species | Country | Locality | Reference |
|---|---|---|---|---|---|---|
| 1 | SIEZC 20201 | MN395604; MN396765 | *Oligodon rostralis* **sp. nov.** | Vietnam | Lam Dong Prov., Bidoup–Nui Ba NP | *this work* |
| 2 | ZMMU R-14304 | MN395601; MN396762 | *Oligodon annamensis* | Vietnam | Dak Lak Prov., Chu Yang Sin NP | *this work* |
| 3 | CBC 01899 | MN395602; MN396763 | *Oligodon annamensis* | Cambodia | Pursat Prov., Veal Veng, Samkos WS | *this work* |
| 4 | ZMMU R-13364 | MN395603; MN396764 | *Oligodon lacroixi* | Vietnam | Phu Tho Prov., Xuan Son NP | *this work* |
| 5 | UMMZ201913 | HM591519 | *Oligodon octolineatus* | Brunei | Tutong Dist., 3 km E of Tutong | *Green, Orlov & Murphy (2010)* |
| 6 | ROM 35626 | HM591526 | *Oligodon chinensis* | Vietnam | Cao Bang Prov., Quang Thanh | *Green, Orlov & Murphy (2010)* |
| 7 | ROM 30970 | HM591528 | *Oligodon chinensis* | Vietnam | Nghe An Prov., 24 km W of Con Cuong | *Green, Orlov & Murphy (2010)* |
| 8 | ROM 34540 | HM591527 | *Oligodon chinensis* | Vietnam | Hai Duong Prov., Chi Linh | *Green, Orlov & Murphy (2010)* |
| 9 | ROM 31032 | HM591524 | *Oligodon chinensis* | Vietnam | Vinh Phuc Prov., Tam Dao NP | *Green, Orlov & Murphy (2010)* |
| 10 | ROM30824 | HM591525 | *Oligodon chinensis* | Vietnam | Tuyen Quang Prov., Pac Ban | *Green, Orlov & Murphy (2010)* |
| 11 | ROM 30823 | HM591529 | *Oligodon formosanus* | Vietnam | Tuyen Quang Prov., Pac Ban | *Green, Orlov & Murphy (2010)* |
| 12 | ROM30826 | HM591530 | *Oligodon formosanus* | Vietnam | Vinh Phuc Prov., Tam Dao NP | *Green, Orlov & Murphy (2010)* |
| 13 | ROM30939 | HM591531 | *Oligodon formosanus* | Vietnam | Cao Bang Prov., Ba Be | *Green, Orlov & Murphy (2010)* |
| 14 | ROM35629 | HM591533 | *Oligodon formosanus* | Vietnam | Cao Bang Prov., Quang Thanh | *Green, Orlov & Murphy (2010)* |
| 15 | ROM35806 | HM591532 | *Oligodon formosanus* | Vietnam | Hai Duong Prov., Chi Linh | *Green, Orlov & Murphy (2010)* |
| 16 | ROM32261 | HM591534 | *Oligodon ocellatus* | Vietnam | Dak Lak Prov., Yok Don NP | *Green, Orlov & Murphy (2010)* |
| 17 | ROM32260 | HM591521 | *Oligodon taeniatus* | Vietnam | Dak Lak Prov., Yok Don NP | *Green, Orlov & Murphy (2010)* |
| 18 | ROM37091 | HM591522 | *Oligodon taeniatus* | Vietnam | Dong Nai Prov., Cat Tien NP | *Green, Orlov & Murphy (2010)* |
| 19 | ROM32464 | HM591523 | *Oligodon barroni* | Vietnam | Gai Lai Prov., Krong Pa | *Green, Orlov & Murphy (2010)* |
| 20 | USNM520625 | HM591520 | *Oligodon cf. taeniatus* | Myanmar | Chatthin, 2 km WNW Chatthin WS | *Green, Orlov & Murphy (2010)* |
| 21 | CAS204963 | HM591535 | *Oligodon cyclurus* | Myanmar | Ayeyarwady Div., Mwe Hauk | *Green, Orlov & Murphy (2010)* |
| 22 | CAS215636 | HM591536 | *Oligodon cyclurus* | Myanmar | Sagging Div., Alaungdaw Kathapa NP | *Green, Orlov & Murphy (2010)* |
| 23 | ROM37092 | HM591504 | *Oligodon cinereus* | Vietnam | Dong Nai Prov., Cat Tien NP | *Green, Orlov & Murphy (2010)* |
| 24 | CAS213379 | HM591506 | *Oligodon cf. cinereus* | Myanmar | Yangon Div., Hlaw Ga NP | *Green, Orlov & Murphy (2010)* |
| 25 | CAS205028 | HM591507 | *Oligodon cf. cinereus* | Myanmar | Rakhine St., Rakhine Yoma Mts. | *Green, Orlov & Murphy (2010)* |
| 26 | ROM32462 | HM591501 | *Oligodon cinereus* | Vietnam | Hai Duong Prov., Chi Linh | *Green, Orlov & Murphy (2010)* |
| 27 | ROM29552 | HM591502 | *Oligodon cinereus* | Vietnam | Vinh Phuc Prov., Tam Dao NP | *Green, Orlov & Murphy (2010)* |
| 28 | ROM30969 | HM591503 | *Oligodon cinereus* | Vietnam | Nghe An Prov., 24km W of Con Cuong | *Green, Orlov & Murphy (2010)* |
| 29 | CAS215261 | HM591508 | *Oligodon cf. cinereus* | Myanmar | Shan St., Kalaw | *Green, Orlov & Murphy (2010)* |
| 30 | CAS204855 | HM591509 | *Oligodon splendidus* | Myanmar | Mandalay Div., Kyauk Se | *Green, Orlov & Murphy (2010)* |
| 31 | USNM520626 | HM591510 | *Oligodon splendidus* | Myanmar | Chatthin, 2 km WNW Chatthin WS | *Green, Orlov & Murphy (2010)* |
| 32 | CAS210693 | HM591512 | *Oligodon torquatus* | Myanmar | Magwe Div., Pakokku | *Green, Orlov & Murphy (2010)* |
| 33 | CAS215976 | HM591513 | *Oligodon torquatus* | Myanmar | Mandalay Div., Min Gone Taung WS | *Green, Orlov & Murphy (2010)* |
| 34 | CAS213822 | HM591514 | *Oligodon planiceps* | Myanmar | Magwe Div., Shwe Set Taw WS | *Green, Orlov & Murphy (2010)* |
| 35 | CAS210710 | HM591515 | *Oligodon theobaldi* | Myanmar | Mandalay Div., Naung U | *Green, Orlov & Murphy (2010)* |

**Table 2** (*continued*)

| No. | Sample ID | Genbank AN | Species | Country | Locality | Reference |
|---|---|---|---|---|---|---|
| 36 | CAS213896 | HM591516 | *Oligodon theobaldi* | Myanmar | Magwe Div., Shwe Set Taw WS | *Green, Orlov & Murphy (2010)* |
| 37 | CAS213271 | HM591517 | *Oligodon cruentatus* | Myanmar | Yangon Div., Hlaw Ga NP | *Green, Orlov & Murphy (2010)* |
| 38 | ROM27049 | HM591518 | *Oligodon eberhardti* | Vietnam | Cao Bang Prov., Quang Thanh | *Green, Orlov & Murphy (2010)* |
| 39 | TNHC59846 | HM591511 | *Oligodon maculatus* | Philippines | Mindanao, Barangay Baracatan | *Green, Orlov & Murphy (2010)* |
| 40 | RS-OC | KC347328; KC347366 | *Oligodon calamarius* | Sri Lanka | Kandy Dist. | *Pyron et al. (2013)* |
| 41 | RAP 504 | KC347329; KC347367 | *Oligodon sublineatus* | Sri Lanka | Kandy Dist. | *Pyron et al. (2013)* |
| 42 | RAP 483 | KC347327; KC347365 | *Oligodon arnensis* | Sri Lanka | Hambantota Dist. | *Pyron et al. (2013)* |
| 43 | RS 136 | KC347330; KC347368 | *Oligodon taeniolatus* | Sri Lanka | Polonnaruwa Dist. | *Pyron et al. (2013)* |
| **Outgroups:** | | | | | | |
| 44 | ROM23440 | KX694641 | *Eirenis modestus* | — | — | *Alencar et al. (2016)* |
| 45 | SPM002589 | KX909261; HQ267796 | *Lytorhynchus diadema* | — | — | *Tamar et al. (2016)* |
| 46 | ELI509 | MH756319 | *Thrasops jacksonii* | — | — | *Engelbrecht et al. (2019)* |
| 47 | KU290488 | KX660250 | *Philothamnus irregularis* | — | — | *Figueroa et al. (2016)* |
| 48 | — | KJ719252 | *Stichophanes ningshaanensis* | — | — | — |
| 49 | — | GQ181130 | *Oreocryptophis poryphyraceus* | — | — | — |
| 50 | — | KF148620 | *Ptyas major* | — | — | — |
| 51 | — | KF148622 | *Lycodon rufozonatus* | — | — | — |
| 52 | — | KP684155 | *Hebius vibakari* | — | — | — |
The matrilineal genealogy was inferred using Bayesian inference (BI) and Maximum Likelihood (ML) approaches. The best-fitting model for both BI and ML analyses for 12S–16S rRNA fragments was the GTR+G+I model as of DNA evolution suggested by the Akaike Information Criterion (AIC); for cyt *b* gene AIC suggested GTR+G model for first and third codon partitions, and HKY+G+I for second codon partition. BI was conducted in MrBayes 3.1.2 (*Ronquist & Huelsenbeck, 2003*); Metropolis-coupled Markov chain Monte Carlo (MCMCMC) analyses were performed run with one cold chain and three heated chains for twenty million generations and sampled every 2000 generations. Five independent MCMCMC run iterations were performed and 1000 trees were discarded as burn-in. The convergence of the runs were checked by exploring examining the likelihood plots in TRACER v1.6 (*Rambaut et al., 2014*); the effective sample sizes (ESS) were all above 200. Nodal support was assessed by calculating posterior probabilities (BI PP).

ML was conducted using the RAxML web server (http://embnet.vital-it.ch/raxml-bb/; *Kozlov et al., 2018*). Confidence in nodal topology was estimated by non-parametric bootstrapping (ML BS) with 1000 pseudoreplicates (*Felsenstein, 1985*).

We a priori regarded tree nodes with BI PP values over 0.95 and ML BS values 75% or greater as sufficiently resolved; while BI PP values between 0.95 and 0.90 and ML BS values between 75% and 50% were regarded as tendencies. Lower values were regarded as indicating unresolved nodes (*Huelsenbeck & Hillis, 1993*).

## RESULTS

### Phylogenetic relationships of *Oligodon*

**Sequence and statistics.** The final concatenated alignment of the 12S rRNA –16S rRNA fragment and cyt *b* gene sequences contained 3131 base pairs, of which, 1959 sites were conserved and 1049 sites were variable, of which 713 were found to be parsimony-informative. The transition–transversion bias (R) was estimated as 1.89. Nucleotide frequencies were 38.0% (A), 22.0% (T), 24.5% (C), and 15.4% (G) (all data given for ingroup only).

**MtDNA-based genealogy.** Our mtDNA-based genealogy for the genus *Oligodon* (Fig. 2) inferred the following set of phylogenetic relationships, which is generally consistent with the results of *Green, Orlov & Murphy (2010)*. Several well-supported clades were recovered within *Oligodon* (see Fig. 2):

1. Clade 1: Indian and Sri Lankan species (*O. taeniolatus*, *O. calamarius*, *O. sublineatus*; 1.0/100; hereafter node support values are given for BI PP/ML BS, respectively); *O. arnensis* from the same region tends to group with this clade, however with no node support (0.52/-).
2. Clade 2: Species from northern Vietnam (*O. lacroixi* and *O. eberhardti*) (1.0/100).
3. Clade 3: Joining the *O. cinereus* species group (Indochina and Myanmar), and some taxa from Myanmar (*O. splendidus*, *O. theobaldi*, *O. cruentatus*, *O. torquatus*, *O. planiceps*) and Philippines (*O. maculatus*) (1.0/100).
4. Clade 4: Joining other species of *Oligodon* from Indochina and southern China, clustered in the *O. taeniatus* species group (*O. taeniatus* and *O. barroni*; 1.0/98) and

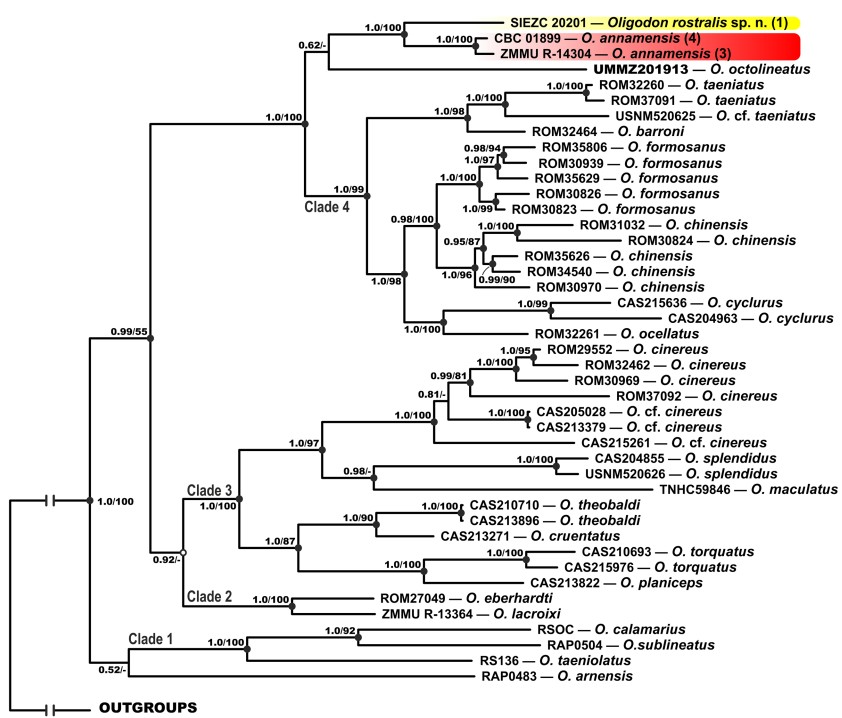

**Figure 2** **Bayesian inference tree of *Oligodon* derived from the analysis of 3,131 bp of 12S rRNA, 16S rRNA and cyt *b* mitochondrial DNA gene sequences.** For voucher specimen information and GenBank accession numbers see Table 1. Numbers at tree nodes correspond to BI PP/ML BS support values, respectively; n-dash denotes no support. Outgroup taxa not shown. Colors of clades and locality numbers correspond to those in Fig. 1.

the *O. cyclurus* species group (*O. cyclurus*, *O. formosanus*, *O. chinensis* and *O. ocellatus*; 1.0/98).

5. The newly discovered *Oligodon* sp. from Bidoup–Nui Ba NP is reconstructed as a sister lineage with respect to two specimens of *O. annamensis* from Vietnam and Cambodia (1.0/100); *O. octolineatus* from Sundaland tends to group with this clade, however with no node support (0.62/-). All these species are clustered together with Clade 4 with strong support (1.0/100) (see Fig. 2).

**Sequence divergence.** The uncorrected *p*-distances for the 16S rRNA gene fragment among and within examined *Oligodon* species are presented in Table 3. Intraspecific distances varied significantly and ranged from $p = 0\%$ in a number of examined species to $p = 2.3\%$ in the *O. cinereus* complex and $p = 2.8\%$ in the *O. cyclurus* complex, what is most likely explained by incomplete taxonomy of these groups (*Green, Orlov & Murphy, 2010*; *David, Das & Vogel, 2008*; *David et al., 2012*); a more detailed study including topotype materials on these species complexes is required.

The interspecific genetic distances within examined *Oligodon* varied from $p = 1.8\%$ (between *O. chinensis* and *O. formosanus*) to $p = 8.5\%$ (between *O. maculatus* and *O. octolineatus*) (Table 3). The newly discovered *Oligodon* sp. lineage from Bidoup–Nui Ba NP is highly divergent from other congeners and is most closely related to *O. annamensis* with

Nguyen et al. (2020), *PeerJ*, DOI 10.7717/peerj.8332

**Table 3 Genetic differentiation of *Oligodon*.** Uncorrected p-distance (percentage) between the sequences of 16S rRNA mtDNA gene (below the diagonal), estimate errors (above the diagonal) and intraspecific genetic *p*-distance (on the diagonal) of *Oligodon* species included in phylogenetic analyses.

| No. | Species | 1 | 2 | 3 | 4 | 5 | 6 | 7 | 8 | 9 | 10 | 11 | 12 | 13 | 14 | 15 | 16 | 17 | 18 | 19 | 20 | 21 | 22 |
|---|---|---|---|---|---|---|---|---|---|---|---|---|---|---|---|---|---|---|---|---|---|---|---|
| 1 | *Oligodon rostralis* **sp. nov.** | — | 0.8 | 1.0 | 0.8 | 1.0 | 0.9 | 0.9 | 1.1 | 1.0 | 1.2 | 1.2 | 1.2 | 1.1 | 1.1 | 1.1 | 1.1 | 1.3 | 1.1 | 1.2 | 1.2 | 1.2 | 1.1 |
| 2 | *O. annamensis* | 3.3 | **0.9** | 1.1 | 0.8 | 1.0 | 0.9 | 1.0 | 1.0 | 1.0 | 1.0 | 1.2 | 1.0 | 1.0 | 1.1 | 0.9 | 1.2 | 1.0 | 1.0 | 1.1 | 1.0 | 1.2 | 1.0 |
| 3 | *O. octolineatus* | 4.4 | 4.9 | — | 1.0 | 1.1 | 1.0 | 1.1 | 1.3 | 1.2 | 1.3 | 1.4 | 1.4 | 1.3 | 1.2 | 1.3 | 1.3 | 1.4 | 1.3 | 1.3 | 1.3 | 1.5 | 1.3 |
| 4 | *O. taeniatus* complex | 3.9 | 3.7 | 5.5 | **0.0** | 0.7 | 0.8 | 0.8 | 0.6 | 0.9 | 1.0 | 1.1 | 1.1 | 0.9 | 0.9 | 0.9 | 1.1 | 1.1 | 1.1 | 1.0 | 1.1 | 1.1 | 1.1 |
| 5 | *O. barroni* | 5.0 | 5.0 | 6.1 | 2.4 | — | 0.9 | 0.9 | 0.9 | 1.1 | 1.2 | 1.2 | 1.2 | 1.1 | 1.1 | 1.2 | 1.2 | 1.3 | 1.2 | 1.1 | 1.2 | 1.3 | 1.3 |
| 6 | *O. formosanus* | 3.8 | 4.1 | 5.8 | 3.2 | 4.3 | **0.5** | 0.5 | 1.0 | 0.9 | 1.0 | 1.1 | 1.1 | 0.9 | 1.0 | 1.0 | 1.0 | 1.1 | 0.9 | 1.1 | 1.1 | 1.2 | 1.0 |
| 7 | *O. chinensis* | 4.5 | 5.0 | 6.5 | 3.2 | 4.3 | 1.8 | **0.7** | 0.9 | 0.9 | 1.1 | 1.1 | 1.1 | 1.0 | 1.0 | 1.1 | 1.1 | 1.1 | 1.0 | 1.1 | 1.2 | 1.3 | 1.1 |
| 8 | *O. cyclurus* complex | 5.8 | 5.1 | 7.1 | 2.9 | 4.7 | 5.4 | 5.5 | **2.8** | 0.9 | 1.0 | 1.1 | 1.1 | 1.0 | 1.0 | 1.0 | 1.1 | 1.1 | 1.1 | 1.1 | 1.1 | 1.2 | 1.2 |
| 9 | *O. ocellatus* | 4.8 | 4.8 | 5.7 | 3.9 | 5.0 | 3.9 | 4.1 | 3.7 | — | 1.1 | 1.2 | 1.2 | 1.1 | 1.1 | 1.2 | 1.1 | 1.2 | 1.2 | 1.3 | 1.2 | 1.3 | 1.1 |
| 10 | *O. cinereus* complex | 6.8 | 6.1 | 7.7 | 6.0 | 7.1 | 6.2 | 6.2 | 6.1 | 6.3 | **2.3** | 0.8 | 0.8 | 0.7 | 0.7 | 0.9 | 0.8 | 0.9 | 0.9 | 0.9 | 0.9 | 1.0 | 1.0 |
| 11 | *O. splendidus* | 7.2 | 7.0 | 8.1 | 6.1 | 7.2 | 6.7 | 6.2 | 6.9 | 7.0 | 4.0 | **0.4** | 0.8 | 0.8 | 0.8 | 0.9 | 0.8 | 1.1 | 1.0 | 1.0 | 1.0 | 1.0 | 1.1 |
| 12 | *O. maculatus* | 7.6 | 6.5 | 8.5 | 6.3 | 7.6 | 6.0 | 6.0 | 7.1 | 7.0 | 4.1 | 3.5 | — | 0.6 | 0.7 | 0.7 | 0.8 | 1.0 | 1.0 | 0.9 | 0.9 | 0.9 | 1.0 |
| 13 | *O. theobaldi* | 5.2 | 4.7 | 6.6 | 3.9 | 5.2 | 4.5 | 5.0 | 5.1 | 5.4 | 3.7 | 3.7 | 2.8 | **0.0** | 0.4 | 0.6 | 0.7 | 0.9 | 0.8 | 0.9 | 1.0 | 0.9 | 0.9 |
| 14 | *O. cruentatus* | 5.9 | 5.4 | 7.2 | 4.6 | 5.9 | 5.1 | 5.6 | 5.6 | 6.1 | 4.3 | 4.4 | 3.5 | 0.7 | — | 0.7 | 0.8 | 1.0 | 0.8 | 1.0 | 1.0 | 0.9 | 1.0 |
| 15 | *O. torquatus* | 6.5 | 5.5 | 8.2 | 5.1 | 6.2 | 5.7 | 6.2 | 6.2 | 6.8 | 4.9 | 5.0 | 4.1 | 2.3 | 2.9 | **1.3** | 0.7 | 1.0 | 0.9 | 0.9 | 0.9 | 1.0 | 0.9 |
| 16 | *O. planiceps* | 6.3 | 6.2 | 7.9 | 5.7 | 6.8 | 5.2 | 5.9 | 6.7 | 6.3 | 5.0 | 5.2 | 4.6 | 2.4 | 3.1 | 3.2 | — | 1.1 | 1.0 | 1.0 | 1.0 | 1.0 | 1.2 |
| 17 | *O. eberhardti* | 6.6 | 5.9 | 7.5 | 6.4 | 7.7 | 6.3 | 6.1 | 6.6 | 6.6 | 5.4 | 6.6 | 5.3 | 4.4 | 5.1 | 5.4 | 6.4 | — | 0.7 | 1.1 | 1.0 | 1.2 | 1.0 |
| 18 | *O. lacroixi* | 5.2 | 4.8 | 7.0 | 5.0 | 6.3 | 4.4 | 5.2 | 5.9 | 5.9 | 4.9 | 6.3 | 5.4 | 3.3 | 3.9 | 4.7 | 5.2 | 2.2 | — | 0.9 | 1.0 | 1.0 | 0.9 |
| 19 | *O. calamarius* | 6.1 | 6.0 | 7.2 | 5.9 | 7.0 | 5.5 | 6.4 | 6.7 | 7.2 | 5.7 | 5.7 | 5.2 | 5.0 | 5.7 | 6.4 | 5.4 | 6.6 | 5.4 | — | 0.8 | 0.9 | 1.0 |
| 20 | *O. sublineatus* | 6.6 | 5.8 | 7.2 | 6.1 | 7.2 | 5.9 | 6.4 | 6.8 | 7.0 | 5.3 | 5.0 | 4.1 | 4.8 | 5.5 | 5.8 | 5.7 | 5.7 | 5.5 | 3.5 | — | 0.9 | 1.0 |
| 21 | *O. taeniolatus* | 6.5 | 6.7 | 7.9 | 6.1 | 7.4 | 6.1 | 6.6 | 7.1 | 7.4 | 5.4 | 5.2 | 4.4 | 3.5 | 4.1 | 5.7 | 5.0 | 6.2 | 5.4 | 3.7 | 3.3 | — | 1.1 |
| 22 | *O. arnensis* | 5.7 | 5.1 | 7.7 | 5.7 | 7.0 | 4.7 | 5.6 | 7.0 | 6.6 | 6.4 | 6.7 | 5.5 | 4.6 | 5.2 | 5.6 | 6.3 | 6.2 | 4.8 | 5.9 | 5.7 | 5.7 | — |

$p = 3.3\%$ of sequence divergence in 16S rRNA gene between these taxa. This divergence value is notably higher than the genetic differentiation between many other recognized *Oligodon* species (see Table 3), thus suggesting that the divergence between *Oligodon* sp. and *O. annamensis* likely reached species status. Genetic divergence between Vietnamese and Cambodian populations of *O. annamensis* is minimal and comprised $p = 0.9\%$ of substitutions (Table 3).

## Systematics

Our mtDNA-genealogy of *Oligodon* demonstrated that *Oligodon* sp. from Bidoup–Nui Ba NP represents a new previously unknown lineage, sister to *O. annamensis*; both species are clustered with the *O. taeniatus* and *O. cyclurus* species groups with strong support. Though genetic divergence between Cambodian and Vietnamese populations of *O. annamensis*, separated from each other by over 600 km distance, is small ($p = 0.9\%$); genetic differentiation between *Oligodon* sp. from Bidoup–Nui Ba NP and *O. annamensis* is much higher ($p = 3.3\%$) and reaches species-level (see Table 3). We thus confirm the identification of the Cambodian population as *O. annamensis* (previously described by *Neang & Hun, 2013*), and also provide a morphological analysis of all presently known specimens of *O. annamensis* (see Table 4). Our results are further corroborated by our morphological analysis (see below), which uncovered significant differences between *Oligodon* sp. from Bidoup–Nui Ba NP, *O. annamensis* and other congeners. These results support our hypothesis that this recently discovered lineage of *Oligodon* represents an undescribed species, which we describe below:

### *Oligodon rostralis* sp. nov.

(Figures 3–7; Tables 4–5)

**Holotype.** SIEZC 20201, adult male from Bidoup–Nui Ba National Park, ca. 6 km northwards from Da Nhim village, Da Chais Commune, Lac Duong District, Lam Dong Province, southern Vietnam (12.1518°N, 108.5279°E; elevation 1,622 m a.s.l.), collected on a steep slope near to mountain summit in montane evergreen pine forest by Bang Van Tran and Linh Hoang Nguyen at 23 h on June 13, 2017.

**Diagnosis.** The new species is assigned to the genus *Oligodon* Fitzinger, 1826 on the basis of its phylogenetic position and the following morphological attributes: posterior maxillary teeth enlarged and compressed; head short, barely distinct from neck; eye well-developed with round pupil; rostral enlarged; body cylindrical with smooth scales; ventrals rounded; subcaudals paired. *Oligodon rostralis* **sp. nov.** is distinguished from its congeners by a combination of the following morphological characters: medium size in adults (male TL = 582 mm); head small and broad with long largely protruding snout; 15 dorsal scale rows at neck and midbody and 13 rows before vent; ventrals 167, subcaudals 47 in male; single preocular, single postocular; loreal and presubocular absent; six supralabials, third and fourth entering orbit; six infralabials, anterior four contacting chin shields; internasals separate from prefrontals; nasal divided; single anterior and two posterior temporals; cloacal plate undivided; comparatively short hemipenis, forked in anterior one third of their length, extending to 8th subcaudal, lacking spines and papillae, bearing prominent

Table 4 **Morphological data on *Oligodon rostralis* sp. nov. and all known specimens of *O. annamensis*.** Measurements in mm; for abbreviations see Materials and methods section.

| Museum ID | Sex | SVL | TaL | TL | RTL | HL | HW | HW/HL | SnL | EyeL | FrL | FrW | IOD | IND | DEN | ASR | MSR | PSR | RED 1 |
|---|---|---|---|---|---|---|---|---|---|---|---|---|---|---|---|---|---|---|---|
| *Oligodon rostralis* **sp. nov.** | | | | | | | | | | | | | | | | | | | |
| SIEZC 20201* | m | 468 | 114 | 582 | 19.6% | 14.5 | 10.6 | 73.2% | 6.3 | 2.0 | 5.4 | 4.5 | 6.4 | 3.6 | 6 | 15 | 15 | 13 | 113 |
| *Oligodon annamensis* | | | | | | | | | | | | | | | | | | | |
| ZMMU R-14304 | m | 331 | 81 | 412 | 19.7% | 13.2 | 7.3 | 55.1% | 4.2 | 1.8 | 4.0 | 3.4 | 4.8 | 3.6 | 8 | 13 | 13 | 13 | – |
| CBC 01899 | m | 152 | 35 | 187 | 18.7% | 9.7 | 5.2 | 53.6% | 3.0 | 1.7 | 3.1 | 2.4 | 3.6 | 2.4 | 7 | 13 | 13 | 13 | – |
| MNHN 8815 | m | 111 | 22 | 133 | 16.6% | 8.0 | 4.5 | 56.3% | 2.5 | 1.4 | 3.0 | 2.5 | 3.5 | 2.3 | 7-8 | 13 | 13 | 13 | – |
| USNM 90408* | f | 220 | 29 | 249 | 11.7% | 9.4 | 5.8 | 61.7% | 3.0 | 1.5 | 3.3 | 2.6 | 4.1 | 2.3 | 8 | 13 | 13 | 13 | – |

| Museum ID | VS | SC | Total Sc. | AP | LOR | SL | SL-eye | IL | NAS | IL-contact | IL-CS | PrO | PrsOc | PtO | Ate | Pte |
|---|---|---|---|---|---|---|---|---|---|---|---|---|---|---|---|---|
| *Oligodon rostralis* **sp. nov.** | | | | | | | | | | | | | | | | |
| SIEZC 20201* | 167 | 47 | 214 | 1 | 0 | 6 | 3-4 | 6 | D | 1 | 1-4 | 1 | 0 | 1 | 1 | 2 |
| *Oligodon annamensis* | | | | | | | | | | | | | | | | |
| ZMMU R-14304 | 157 | 43 | 200 | 1 | 0 | 6 | 3-4 | 6 | E | 1 | 1-4 | 1 | 0 | 1 | 1 | 2 |
| CBC 01899 | 148 | 46 | 194 | 1 | 0 | 6/5 | 3-4/2-3 | 6 | E | 1 | 1-3 | 1 | 0 | 1 | 1 | 1 |
| MNHN 8815 | 146+2 | 46 | 192 | 1 | 0 | 6 | 3-4 | 6 | E | 1 | 1-4 | 1 | 0 | 1 | 1 | 2 |
| USNM 90408* | 170 | 30 | 200 | 1 | 0 | 6 | 3-4 | 6 | E | 1 | 1-4 | 1 | 0 | 1 | 1 | 2 |

| Museum ID | Hemipenis status | Hemipenis shape | Hemipenis ornamentation | Hemipenis length | Body color | Color pattern | Body blotches | Tail blotches | Stripes | Venter | Reference |
|---|---|---|---|---|---|---|---|---|---|---|---|
| *Oligodon rostralis* **sp. nov.** | | | | | | | | | | | |
| SIEZC 20201* | right everted | 1/3 forked, no papillae | flounced, no spines | reaching 8th sub-caudal | dorsum and tail greyish-brown | middorsal light stripe and dark blotches | 18 large dark blotches | 4 dark blotches | light-orange middorsal stripe with indistinct borders | venter and tail underside pale-cream, with intermittent black quadrangular spots; tail in life light orange | *this paper* |
| *Oligodon annamensis* | | | | | | | | | | | |
| SIEZC 20201* | right everted | 1/3 forked, no papillae | flounced, no spines | reaching 8th sub-caudal | dorsum and tail greyish-brown | middorsal light stripe and dark blotches | 18 large dark blotches | 4 dark blotches | light-orange middorsal stripe with indistinct borders | venter and tail underside pale-cream, with intermittent black quadrangular spots; tail in life light orange | *this paper* |
| *Oligodon annamensis* | | | | | | | | | | | |
| ZMMU R-14304 | left dissected | deeply forked with 2 papillae | transverse ridges, no spines | reaching 20th sub-caudal | dorsum and tail dark brown | light cross bars | 10 faint white cross bars edged with black | 5 beige cross bars | no | coral-red with numerous black bars and qudrangular blotches | *this paper* |
| CBC 01899 | – | – | – | – | dorsum and tail dark brown | light cross bars | 10 orange cross bars 2–3 scales wide | 3 orange cross bars on tail | no | orange with sparse black subrectangular blotches | *Neang & Hun, 2013*; *our data* |
| MNHN 8815 | dissected | deeply forked with 2 papillae | transverse ridges, no spines | reaching 17th sub-caudal | dorsum and tail light brown | light cross bars | 10 white cross bars edged with black | 2 white cross bars | no | white with dark quadranglar spots and bars | *Leviton, 1960*; *our data* |
| USNM 90408* | – | – | – | – | dorsum and tail light brown | light cross bars | 9 white cross bars edged with black | 3 white cross bars | no | white with dark quadranglar spots and bars | *Leviton, 1953*; *our data* |

**Notes.**
*An asterisk (*) denotes the holotype of a species.

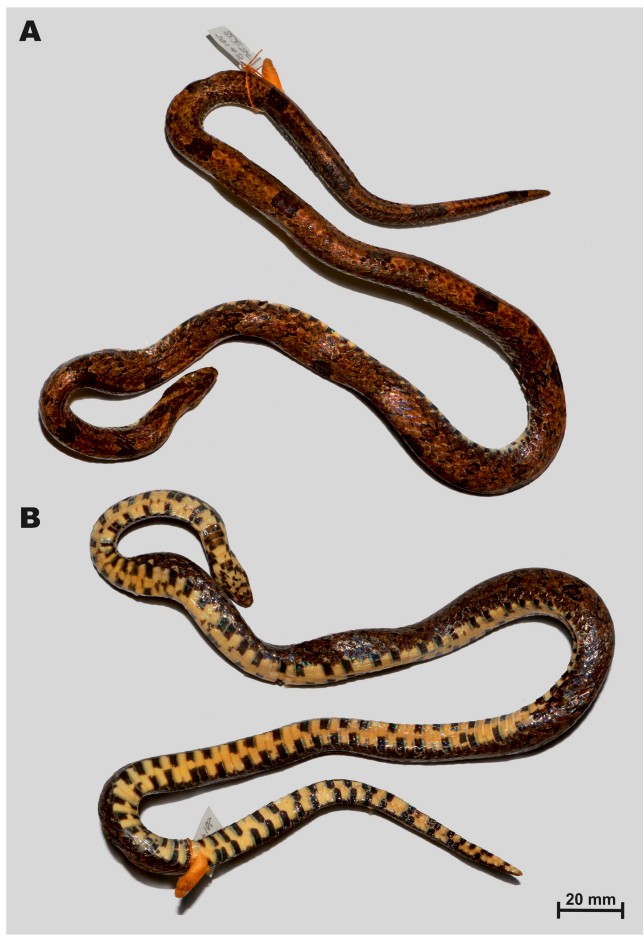

**Figure 3** Holotype of *Oligodon rostralis* sp. nov. in preservative (SIEZC 20201, male) in dorsal (A) and in ventral (B) views. Scale bar denotes 20 mm. Photos by Bang Van Tran.

transverse flounces and distal calyces; six maxillary teeth, the posterior three being enlarged; dark temporal streak present, edged with white; 14+4 large dark-brown dorsal blotches; bright-orange vertebral stripe on tail and dorsum; and ventral surfaces in life cream with quadrangular spots.

**Description of holotype.** Measurements and scale counts of the holotype are presented in Table 4. Adult male of medium size (TL 582 mm), body robust and cylindrical (Fig. 3); SVL 468 mm; head small, comparatively short and wide (HW/HL = 73.2%), ovoid in dorsal view, faintly distinct from the poorly defined neck; tail quite long (19.6% of total length), 114 mm in length; robust, abruptly tapering; eye small, comprising approximately 13.5% of the head length; snout long, protruding (SnL/HL ratio 43.6%); eye diameter much shorter than the distance between eye and nostril; pupil round;

*Body scalation.* Dorsal scales smooth, in 15-15-13 rows, scale row reduction from 15 to 13 at ventral 113; vertebral scales similar to other dorsal scales in size and shape; outermost

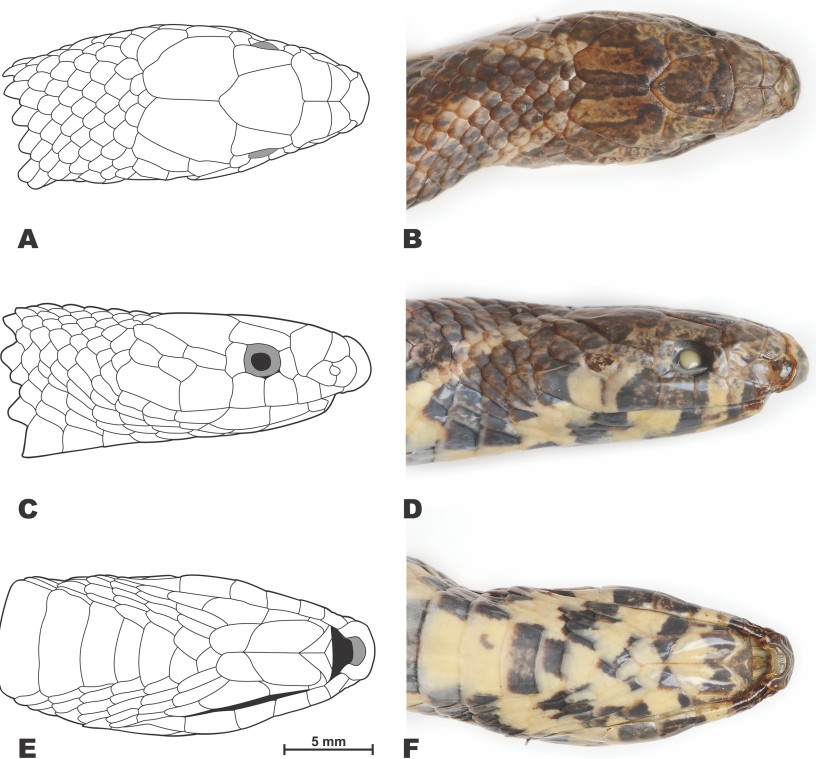

**Figure 4** Drawings (A, C, E) and photos (B, D, F) showing head scalation of the holotype *Oligodon rostralis* **sp. nov. in preservative (SIEZC 20201, male).** (A, B) Dorsal view; (C, D) lateral view; (E, F) ventral view. Scale bar equals 5 mm. Drawings and photos by Linh Hoang Nguyen.

dorsal scales slightly enlarged; 167 ventrals; cloacal plate entire; 47 subcaudals, all paired, terminal caudal scale in a shape of sharply pointed cap (Fig. 3B).

*Head scalation.* Details of head scalation are shown in Fig. 4. From dorsal view (Figs. 4A–4B), head scalation comprising single rostral, two internasals, two prefrontals, two supraoculars, single frontal, and two parietals. Rostral large, thick, wider than high, extending on to the dorsal surface of the snout, visible from above, pointed posteriorly and inserting deeply between internasals, with a deep crease ventrally, contacting nasals, internasals and first supralabial on both sides; the portion of rostral visible from above shorter than its distance from frontal; internasals sub-rectangular, in broad contact, shorter than prefrontals, each contacting rostral, prefrontal, internasal and paired nasals on both sides; prefrontals large, pentagonal, wider than long and larger than internasals, curving dorsolaterally into loreal region, each contacting internasal and posterior portion of nasal anteriorly, second supralabial laterally, and preocular, supraocular and frontal posteriorly; supraoculars subrectangular, elongated, widening posteriorly, approximately half as wide as long, contacting the orbit, preocular and postocular laterally, prefrontal, frontal and parietal medially; frontal large, pentagonal, longer than wide, narrowing posteriorly, posterior angle rather acute, contacting prefrontals, supraoculars and parietals on both

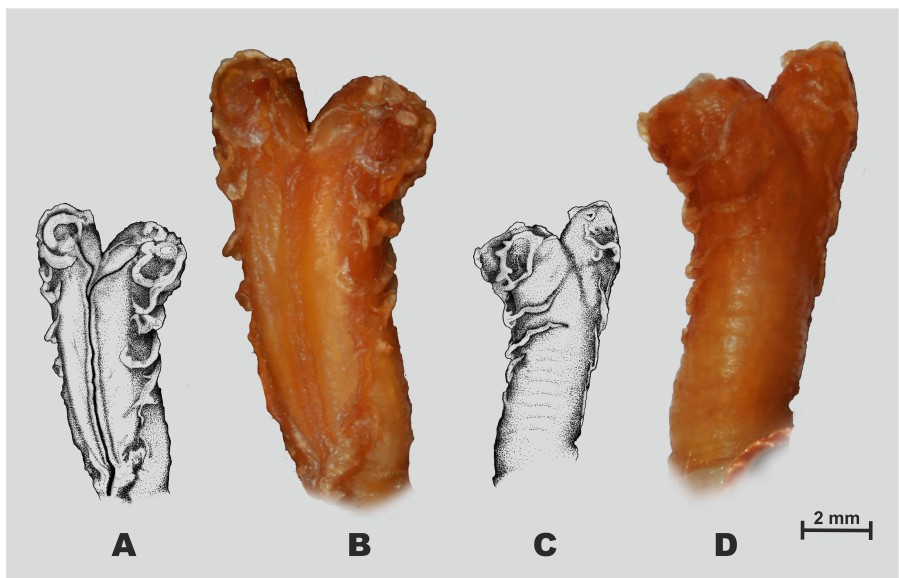

**Figure 5** **Hemipenial morphology of *Oligodon rostralis* sp. nov. holotype in preservative (SIEZC 20201, male).** (A, C) Drawings; (B, D) photos; (A, B) right hemipenis in sulcal view; (C, D) right hemipenis in asulcal view. Scale bar equals 2 mm and represents photograph of hemipenis. Photos by Bang Van Tran; drawings by Platon V. Yushchenko.

sides; parietals irregularly trapeziform, about 1.5 time larger than frontal, anteriorly contacting frontal, supraoculars and postoculars on each side, bordered posteriorly by five small scales and laterally by the first and upper second temporals; no enlarged nuchal scales present.

In lateral view (Figs. 4C–4D), head scalation comprising a sub-rectangular nasal, vertically divided by prominent suture and pierced by large nostril, nasal on each side contacting rostral anteriorly, internasal and prefrontal dorsally, and first two supralabials ventrally; loreal and presubocular scales absent; 1/1 rectangular preocular, notably higher than wide, separated from nasal by the lateral part of the prefrontal, also contacting second and third supralabials ventrally and supraocular dorsally; 1/1 rectangular postocular, almost equal in size to preocular, contacting fourth and fifth supralabials ventrally, anterior temporal and parietal posteriorly and supraocular dorsally; six supralabials: I. the smallest, in contact with nasal, II. in contact with nasal, prefrontal and preocular, III. in contact with preocular and the orbit, IV. in contact with the orbit and postocular, V. in contact with postocular, anterior temporal and lower posterior temporal, VI. in contact with lower posterior temporal and scale dorsally, and with two smaller scales posteriorly, V. and VI. strongly enlarged; supralabial scale size formula: I<II<III=IV<V<VI; 1+2 temporals on each side, the upper ones pentagonal, elongated and narrow, upper posterior temporal slightly larger than the anterior, the lower posterior temporal rhomboid, ca. two times smaller than the upper ones, posteriorly contacting an enlarged scale of same size.

In ventral view (Figs. 4E–4F), 6/6 infralabials: I. in contact with mental anteriorly, in contact with each other medially; anterior three in contact with anterior chin shield; the

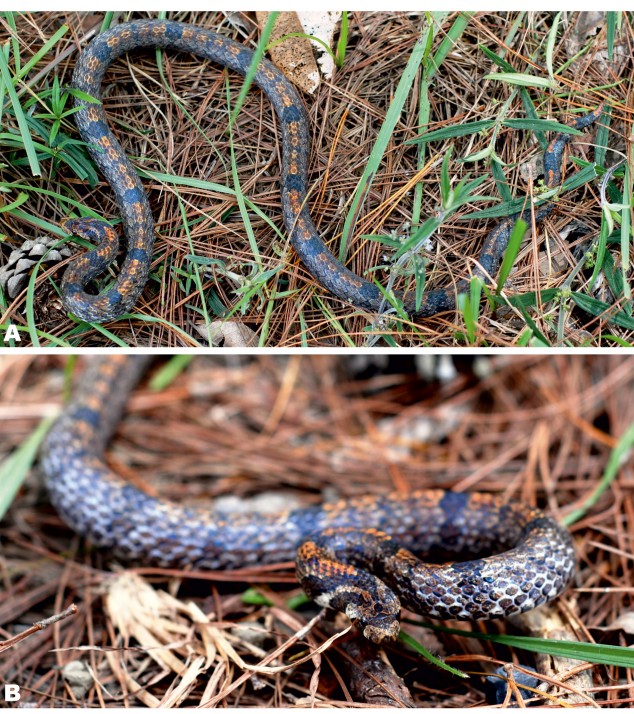

**Figure 6** **Holotype of *Oligodon rostralis* sp. nov. in life *in situ* (SIEZC 20201, male) in dorsal (A) and in frontal (B) views.** Photos by Linh Hoang Nguyen.

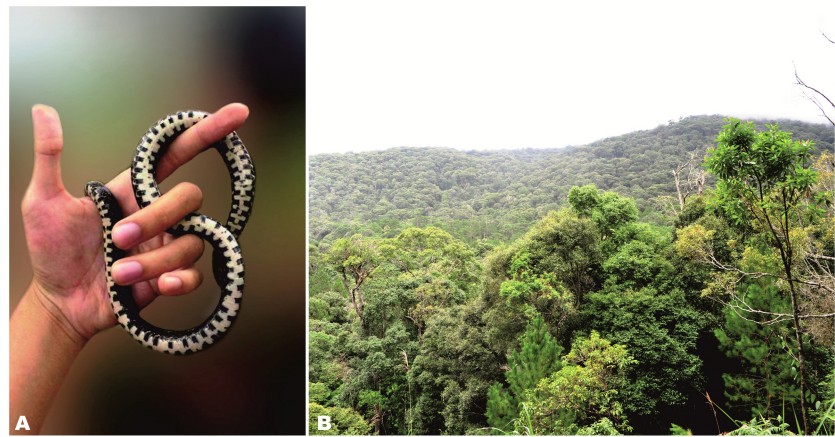

**Figure 7** **Natural habitat of *Oligodon rostralis* sp. nov. at the type locality in pine forest dominated by *Pinus kesiya* Royle ex Gordon in Bidoup–Nui Ba NP, Lam Dong Province, Langbian Plateau, southern Vietnam.** (A) Live ventral coloration of *Oligodon rostralis* **sp. nov.** (SIEZC 20201, male); (B) general view of the macrohabitat (elevation 1,622 m a.s.l.). Photos by Linh Hoang Nguyen.

fourth largest and touching posterior chin shield; mental small, triangular; 2/2 enlarged, elongated chin shields, anterior pair twice as long as the posterior pair; three small gular scales between posterior chin shields and first preventral.

# PeerJ

**Table 5  Comparison of morphological characters of Oligodon rostralis sp. nov. with Indochinese species of Oligodon having 1315 dorsal scale rows (DSR).** The characters and data for other species taken from *Das (2010)*; *Gong et al. (2007)*; *Leviton (1953)*; *Leviton (1960)*; *Neang, Grismer & Daltry (2012)*; *Neang & Hun (2013)*; *Orlov et al. (2010)*; *Pauwels et al. (2002)*; *Pellegrin (1910)*; *Pham, Nguyen & Nguyen (2014)*; *Smith (1943)*; *Taylor (1965)*.

| Species | DSR | SL | SL-E | IL | NAS | InN/PF | PrO | PtO | AP | LOR | Ate | Pte | VS | SC | DEN | RTL |
|---|---|---|---|---|---|---|---|---|---|---|---|---|---|---|---|---|
| *Oligodon rostralis* **sp. nov.** | 15-15-13 | 6 | 3-4 | 6 | D | S | 1 | 1 | E | 0 | 1 | 2 | 167 | 47 | 6 | 19.6 |
| *Oligodon annamensis* | 13-13-13 | 6 | 3-4 | 6 | E | S | 1 | 1 | E | 0 | 1 | 2 | 148-170 | 30-46 | 7-8 | 11.6-19.7 |
| *Oligodon catenatus* | 13-13-13 | 6 | 3-4 | 6 | E | F | 1 | 2 | D | 0 | 1 | 2 | 179-212 | 31-43 | 7 | 12.6-13.3 |
| *Oligodon eberhardti* | 13-13-13 | 6 | 2-3 | 6 | E | F | 1 | 1 | D | 1 | 1 | 2 | 165-187 | 31-40 | ? | 15.1 |
| *Oligodon lacroixi* | 17-15-15 | 5 | 2-3 | 6 | E | F | 1 | 2 | D | 0 | 1 | 2 | 162-178 | 25-34 | 8-12 | 10.5-11.5 |
| *Oligodon inornatus* | 15-15-15 | 8 | 4-5 | 8 | D | S | 1 | 2 | E | 1 | 1 | 2 | 169-174 | 31-43 | 10-11 | 15.5 |
| *Oligodon kampucheaensis* | 15-15-15 | 8 | 4-5 | 8 | D | S | 1 | 2 | E | 1 | 1 | 2 | 165 | 39 | 11 | 15.1 |
| *Oligodon jintakunei* | 15-15-15 | 7 | 3-4 | 7 | PD | F | 1 | 1 | D | 1[*] | 1 | 1 | 189 | 46 | 6 | 17.5 |
| *Oligodon lungshenensis* | 15-15-15 | 6 | 3-4 | 6 | E | F | 1 | 2 | D | 0 | 1 | 2 | 163-179 | 31-38 | 8 | 20.0 |
| *Oligodon ornatus* | 15-15-15 | 6 | 3-4 | 7 | E or D | S | 1-2 | 1-2 | D | 0 | 1 | 2 | 156-182 | 27-44 | 6-8 | 15.6 |
| *Oligodon hamptoni* | 15-15-15 | 5 | 2-3 | 5 | E | F | 1 | 2 | D | 0 or 1[*] | 1 | 2 | 160-175 | 30-32 | 7-8 | 12.7 |
| *Oligodon mcdougalli* | 13-13-13 | 7 | 3-4 | 7 | ? | S | ? | ? | D | 0 | 1 | 2 | 200 | 39 | ? | ? |
| *Oligodon planiceps* | 13-13-13 | 5 | 3 | 6 | E | S | 1 | 2 | D | 1 | 1 | 1 | 132–145 | 22–27 | 10 | 9.6 |
| *Oligodon torquatus* | 15-15-15 | 7 | 3-4 | 7 | E | S | 1 | 2 | D | 1 | 1 | 2 | 144–169 | 25–34 | 15-16 | 11.1 |
| *Oligodon dorsalis* | 15-15-13 | 7 | 3-4 | 7 | D | S | 1 | 1 | D | 1 | 1 | 2 | 162-188 | 27-51 | 6-10 | 16.1-19.3 |
| *Oligodon melaneus* | 15-15-15 | 7 | 3-4/4-5 | 7 | PD | S | 1 | 2 | D | 1[*] | 1 | 2 | 152-160 | 39-40 | 7 | 15.0-16.6 |
| *Oligodon brevicauda* | 15-15-15 | 7 | 3-4 | ? | D | F | 2 | 2 | ? | 0 | 1 | 2 | 158-173 | 25-29 | 7-8 | 9.6-11.0 |
| *Oligodon erythrorhachis* | 15-15-15 | 7 | 3-4 | 8 | E | S | 1 | 2 | D | 0 | 1 | 2 | 154 | 46 | 7-8 | 16.5 |

| Species | Hemipenis | Dorsal pattern | Ventral pattern | Distribution | Distribution in Vietnam |
|---|---|---|---|---|---|
| *Oligodon rostralis* **sp. nov.** | 1/3 forked, flounced, no spines | 14+4 large dark blotches, light middorsal stripe | cream with black quadrangular spots | S Vietnam | Lam Dong |
| *Oligodon annamensis* | deeply forked, transverse ridges and papillae | 10 light narrow crossbars | red with dark bars or quadrangular spots | S Vietnam, Cambodia | Lam Dong, Dak Lak |
| *Oligodon catenatus* | not forked, spinose throughout | 4 longitudinal stripes | red with black quadrangular spots | S China, E India, N Myanmar, N Vietnam, Cambodia (?) | Lao Cai, Vinh Phuc, Son La |
| *Oligodon eberharti* | ? | longitudinal stripes | red with dark bars or quadrangular spots | S China, N Myanmar, N Laos, N Vietnam, Cambodia (?) | Lai Chau, Lao Cai, Tuyen Quang, Cao Bang, Bac Kan, Vinh Phuc, Son La, Ha Tay |
| *Oligodon lacroixi* | not forked | 4 longitudinal stripes | dark bars or spots | S China; Vietnam | Lao Cai, Phu Tho |
| *Oligodon inornatus* | not forked | uniform pale brown | uniform, no dark bars or spots | Cambodia, Thailand, Vietnam (?) | ? |
| *Oligodon kampucheaensis* | deeply bifurcated | 17+3 light narrow crossbars | dark quadrangular spots in posterior half | Cambodia | — |
| *Oligodon jintakunei* | ? | 11+3 light narrow crossbars | uniform, no dark bars or spots | S Thailand | — |
| *Oligodon lungshenensis* | ? | 4 dark longitudinal stripes, 9–12 brown crossbars | orange-red with black quadrangular spots | S China | — |

**Table 5** (*continued*)

| Species | Hemipenis | Dorsal pattern | Ventral pattern | Distribution | Distribution in Vietnam |
|---|---|---|---|---|---|
| *Oligodon ornatus* | not forked, spinose throughout | 7–9+2–3 dark crossbars | orange-red with black quadrangular spots | Taiwan | — |
| *Oligodon hamptoni* | not forked, spinose flounces | light vertebral and two dorsolateral stripes | red with dark bars or quadrangular spots | N Myanmar | — |
| *Oligodon mcdougalli* | ? | black with rusty middorsal stripe | uniform black with light mottling | Myanmar | — |
| *Oligodon planiceps* | not forked, spinose and papillae | pale brown with dark reticulations | uniform yellow | S Myanmar | — |
| *Oligodon torquatus* | not forked, no spines, with folds | 4 longitudinal stripes | white, black spots posteriorly | C Myanmar | — |
| *Oligodon dorsalis* | 1/3 forked, flounced, basal spines | dark brown with light middorsal stripe | unfiorm orange or white with black quadrangular spots | NE India, N Myanmar | — |
| *Oligodon melaneus* | not forked, spinose throughout | blackish-brown with speckles | uniform blue-grey | NE India | — |
| *Oligodon brevicauda* | ? | brown with light vertebral stripe | whitish with black quadrangular spots | S India | — |
| *Oligodon erythrorhachis* | ? | red vertebral stripe with many black crossbars | whitish with black quadrangular spots | NE India | — |

**Notes.**

Abbreviations: DSR, dorsal scale rows; SL, number of supralabials; SL-E, supralabials touching the eye; IL, number of infralabials; NAS, nasal (D - divided, E - entire, PD - partially divided); InN/PF, internasal - prefrontal relationships (S - separate, F - fused); PrO, number of preoculars; PtO, number of postoculars; AP, cloacal plate (E - entire, D - divided); LOR, loreal (0 - absent, 1 - present, * - vestigal); Ate, number of anterior temporals; Pte, number of posterior temporals; VS, number of ventrals; SC, number of subcaudals; DEN, number of maxillary teeth; RTL, relative tail length (in %).

*Dentition.* Maxillary teeth 6, curved posteriorly, smaller and shorter anteriorly; posterior three being notably enlarged, flattened and kukri-shaped.

*Hemipenial morphology.* Right hemipenis was everted prior to preservation and is shown in Fig. 5. Hemipenis rather short, the everted organ hardly reaching 8th subcaudal; bilobed, bifurcating at distal fifth of its length; organ semi-capitate and semi-calyculate; the sulcus spermaticus is bifurcated at around the proximal one-fifth of the hemipenial body and centrolineal along both lobes (Fig. 5A). The sulcal surface of hemipenis is mostly smooth (Fig. 5A), laterally and on asulcal surface hemipenis covered with several fleshy flounces, lacking spines or papillae-like structures (Fig. 5B); distal ends of hemipenial lobes with small indistinct calyces.

*Colouration (in life).* Dorsal ground color (Fig. 6A) dark brownish-gray with dense white reticulation between scales; dorsal pattern consisting of 18 large irregular blackish butterfly-shaped blotches, of which 14 are located on body and 4 on tail, the distances between two blotches comprises ca. 8–10 dorsal scale lengths; a bright orange vertebral stripe running from the base of the head to the tail tip; vertebral stripe width comprising from one to three dorsal scale rows; some dorsal scales edged with dark-brown forming an indistinct speckled or dashed pattern between blotches, lower rows of dorsal scales fringed with white. Dorsal ground color along head is grayish-brown (Fig. 6B), a butterfly-shaped marking with rusty tint with a rounded dark spot located on frontal, three separated dark-brown chevrons (one short between the eyes, forming two dark brown streaks running across the eye to the angle of the mouth; and two longer ones running from frontal postero-ventrally to neck and posteriorly to the base of the head, respectively); throat and ventral underside

pale-cream laterally with alternating quadrangular black spots scattered from throat until tail (Fig. 7A); underside of tail orange-cream.

*Colouration (in preservative).* (Fig. 3), after two years in alcohol, coloration faded but pattern remained unchanged; body brown, vertebral stripe became somewhat dark-orange and less distinct (Fig. 3, A); dorsal blotches and head marking dark brown with blackish margins remained unchanged; throat, venter and tail underside cream-white, black quadrangular spots remained unchanged (Fig. 3B).

**Etymology**. The specific name "*rostralis*" is a Latin adjective in the nominative singular, masculine gender, derived from Latin words "*rostrum*" for "snout" or "beak" in reference to protruding snout distinctive for the new species. We suggest the following common names for the new species: "*Long-snouted kukri snake*" (English), "*Rắn khiêm mõm dài*" (Vietnamese), and "*Dlinnorylyi oligodon*" (Russian).

**Distribution.** At present the new species is known only from the type locality in Bidoup–Nui Ba NP, in the eastern part of Langbian Plateau, southern Vietnam (see Fig. 1, locality 1). This montane area is characterized by high levels of local endemism (*Nazarov et al., 2012*; *Poyarkov et al., 2014*; *Poyarkov et al., 2015a*; *Poyarkov et al., 2015b*; *Poyarkov et al., 2017*; *Poyarkov et al., 2019b*; *Stuart et al., 2011*; *Rowley et al., 2016*); further research is needed to clarify the distribution of the new species.

**Habitat and natural history.** The type specimen was collected on the steep slope close to the mountain summit (Fig. 7), at late night (23 h). The animal was found on ground in leaf litter on the edge of the mixed-pine forest (dominated by *Pinus keysia* Royle ex Gordon) and evergreen montane broadleaf forest (dominated with trees of the families Fabaceae, Fagaceae, and few large pine trees of *Pinus keysia*, with understory consisting mostly of Poaceae –different species of bamboo) (Fig. 7B). In the pine forest, understory is dominated by Fagaceae family while ground is covered mostly by grasses and receives high grazing impact by livestock from the villages nearby. In the type locality the new species was recorded in sympatry with some other species of reptiles, including *Cyrtodactylus bidoupimontis* Nazarov, Poyarkov, Orlov, Phung, Nguyen, Hoang & Ziegler, *Scincella rufocaudata* (Darevsky & Nguyen), and *Pareas hamptoni* (Boulenger).

**Phylogenetic position.** *Oligodon rostralis* **sp. nov.** is suggested as a sister species of *O. annamensis* (Fig. 2), from which it is genetically divergent with p-distance 3.3% in 16S rRNA gene (Table 3). Both species are clustered together with the *O. cyclurus* and *O. taeniatus* species groups (Fig. 2).

**Comparisons.** Morphological diagnostics of species based exclusively on hemipenial morphology is often complicated due to insufficiency of data and certain controversy in describing hemipenis character states in *Oligodon* existing in literature (*Smith, 1943*; *Wagner, 1975*; *Vassilieva, 2015*); scalation and coloration features often might be more useful for species identification (*Pauwels et al., 2002*; *David, Das & Vogel, 2008*; *David et al., 2012*; *Neang, Grismer & Daltry, 2012*; *Nguyen et al., 2016*; *Nguyen et al., 2017*). By having 15-15-13 dorsal scale rows, *Oligodon rostralis* **sp. nov.** can be distinguished from other species inhabiting mainland Southeast Asia having greater number of MSR, namely all members of the *O. cyclurus* species group: *O. cyclurus* (Cantor) (19 or 21); *O. formosanus* (Günther) (19); *O. ocellatus* (Morice) (19); *O. fasciolatus* (Günther) (21 or 23); *O. kheriensis*

Achraji & Ray (19); *O. juglandifer* (Wall) (19); *O. chinensis* (Günther) (17); *O. saintgironsi* David, Vogel & Pauwels (17 or 18); *O. culaochamensis* Nguyen, Nguyen, Nguyen, Phan, Jiang & Murphy (17); *O. condaoensis* Nguyen, Nguyen, Le & Murphy (17); *O. macrurus* (Angel) (17); *O. arenarius* Vassilieva (17) and *O. cattienensis* Vassilieva, Geissler, Galoyan, Poyarkov, Van Devender & Böhme (17); phylogenetic position of the latter two species is unclear.

Similarly, by having 15 MSR the new species can be diagnosed from the members of the *O. taeniatus* species group: *O. taeniatus* (Günther) (19); *O. barroni* (Smith) (17); *O. mouhoti* (Boulenger) (17); *O. pseudotaeniatus* David, Vogel & Van Rooijen (17); *O. moricei* David, Vogel & Van Rooijen (17) and *O. deuvei* David, Vogel & Van Rooijen (17).

Most members of the *O. cinereus* species group, which all are believed to have an unforked hemipenis (vs. bilobed hemipenis in the new species; see *Green, Orlov & Murphy, 2010*), can be also distinguished from *Oligodon rostralis* **sp. nov.** by larger MSR: *O. cinereus* (Günther) (17); *O. nagao* David, Nguyen, Nguyen, Jiang, Chen, Teynié & Ziegler (17); *O. joynsoni* (Smith) (17); *O. saiyok* Sumontha, Kunya, Dangsri & Pauwels (17); *O. huahin* Pauwels, Larsen, Suthanthangjai, David & Sumontha (17), and *O. albocinctus* (Cantor) (19 or 21); another member of the *O. cinereus* species group –*O. inornatus* (Boulenger) has 15 MSR and is compared with the new species below.

Diagnostics of *Oligodon rostralis* **sp. nov.** from other mainland Southeast Asian species of *Oligodon* with 15 or 13 dorsal scale rows appear to be the most pertinent (as the number of MSR may vary between these two values due to the position of the dorsal scale row reduction, see *David et al., 2012*); it is summarized in Table 5. From most species with 15 or 13 MSR, the new species can be distinguished by absence of loreal vs. loreal present in *O. eberhardti* Pellegrin; *O. inornatus*; *O. kampucheaensis* Neang, Grismer & Daltry; *O. jintakunei* Pauwels, Wallach, David, Chanhome (vestigial loreal); *O. planiceps* (Boulenger); *O. torquatus* (Boulenger); *O. dorsalis* (Gray) and *O. melaneus* Wall (vestigial loreal). By presence of an entire cloacal plate *Oligodon rostralis* **sp. nov.** can be diagnosed from those species who have the cloacal plate divided, namely from *O. catenatus* (Blyth), *O. eberhardti*, *O. lacroixi* Angel & Bourret, *O. jintakunei*, *O. lungshenensis* Zheng & Huang, *O. ornatus* Van Denburgh, *O. hamptoni* Boulenger, *O. mcdougalli* Wall, *O. planiceps*, *O. torquatus*, *O. dorsalis*, *O. melaneus*, and *O. erythrorhachis* Wall. By having internasals separate from prefrontals the new species can be readily diagnosed from those *Oligodon* species which have these scales fused, including *O. catenatus*, *O. eberhardti*, *O. lacroixi*, *O. jintakunei*, *O. brevicauda* and *O. hamptoni*. By having a single postocular scale *Oligodon rostralis* **sp. nov.** is distinguished from those species which have two postocular scales: *O. catenatus*, *O. lacroixi*, *O. inornatus*, *O. kampucheaensis*, *O. lungshenensis*, *O. hamptoni*, *O. planiceps*, *O. torquatus*, *O. melaneus*, *O. brevicauda* and *O. erythrorhachis*. The new species can be further distinguished from *O. brevicauda* by having a single preocular scale (vs. 2 preoculars). By having six supralabials the new species can be distinguished from *Oligodon* species with five (*O. lacroixi*, *O. hamptoni*, and *O. planiceps*), seven (*O. jintakunei*, *O. mcdougalli*, *O. torquatus*, *O. dorsalis*, *O. melaneus*, *O. brevicauda* and *O. erythrorhachis*), or eight (*O. inornatus* and *O. kampucheaensis*) supralabials.

Among all congeners *Oligodon rostralis* **sp. nov.** morphologically is most similar to *O. annamensis*, to which this species is also most closely related phylogenetically (see Results). However, the new species can be distinguished from males of *O. annamensis* by the following combination of morphological characters: (1) greater number of dorsal scale rows, DSR formula 15-15-13 (vs. DSR formula 13-13-13 in *O. annamensis*); (2) hemipenis bilobed, lobes bifurcating at distal third of body with flounces and lacking papillae (vs. hemipenis bilobed and elongate, lobes bifurcating proximally with papillae and transverse ridges in *O. annamensis*), (3) nasal vertically divided (vs. nasal entire in *O. annamensis*); (4) generally larger total length, 582 mm (vs. maximal total length 412 mm in *O. annamensis*); (5) generally wider head, HW/HL ratio 73.2% (vs. HW/HL ratio 53.6–56.3% in *O. annamensis* males, and 61.7% in female holotype; see Table 4); (6) generally higher number of subcaudals, 47 (vs. 30–46 in *O. annamensis*); (7) dorsal pattern consisting of large dark butterfly-shaped blotches and a light middorsal orange stripe (vs. white narrow crossbars edged with black and no middorsal stripe in *O. annamensis*); (8) ventral color in life cream-white with black quadrangular spots not forming transverse bars (vs. ventral surfaces in life bright coral-red to bright orange with black quadrangular spots forming transverse bars in *O. annamensis*) (see Tables 4 and 5). Finally, the new species is distinguished from *O. annamensis* by a significant divergence in mtDNA gene sequences (up to 3.3% of substitutions in 16S rRNA gene, see Table 3).

## Additional information on *Oligodon annamensis* Leviton, 1953

*Oligodon annamensis Leviton, 1953* Jour. Washington Acad. Sci., 43(12):422.

Figures 8–11; Table 4.

**Holotype.** USNM 90408, young female from ''Blao, Haut Donai, Annam, French Indo-China'' (today environs of Bao Loc, Lam Dong Prov., Vietnam), collected by E. Poilane on March 11, 1933 (Fig. 8).

**Referred specimens.** Three male specimens, including MNHN 8815, young male from Blao, Haut Donai, Station Agricole, collected by E. Poilane on March 11, 1933 (Fig. 9). MNHN 8815, young male from ''Blao, Haut Donai, Station Agricole'', collected by E. Poilane on March 11, 1933 (Fig. 9); ZMMU R-14304, adult male from Chu Pan Phan Mt., Chu Yang Sin NP, Khue Ngoc Dien Comm., Krong Bong Dist., Dak Lak Prov., Vietnam (12.3950°N, 108.3503°E; 1050 m a.s.l.), collected by N.A. Poyarkov on April 14, 2012, described herein (Fig. 10); and CBC 01899, young male from Phnom Samkos WS, Pursat Prov., Cambodia (12.1690°N, 102.9721°E; 916 m a.s.l.), collected by Hun Seiha on April 26, 2012, 1933 (Fig. 11).

**Revised diagnosis.** An *Oligodon* with medium body size in adults (adult male TL up to 412 mm); head small, comparatively narrow, snout not protruding; 13 dorsal scale rows at neck, midbody and before vent; ventrals 146–157, subcaudals 43–46 in males; ventrals 170, subcaudal 30 in female; single preocular, single postocular; loreal and presubocular absent; generally six supralabials, third and fourth entering orbit; six infralabials, anterior three or four contacting the first chin shield; internasals separate from prefrontals; nasal entire; single anterior and one or two posterior temporals; cloacal plate entire; hemipenis deeply bilobed, bearing two long and thin papillae, reaching the 20th subcaudal; 7–8 maxillary

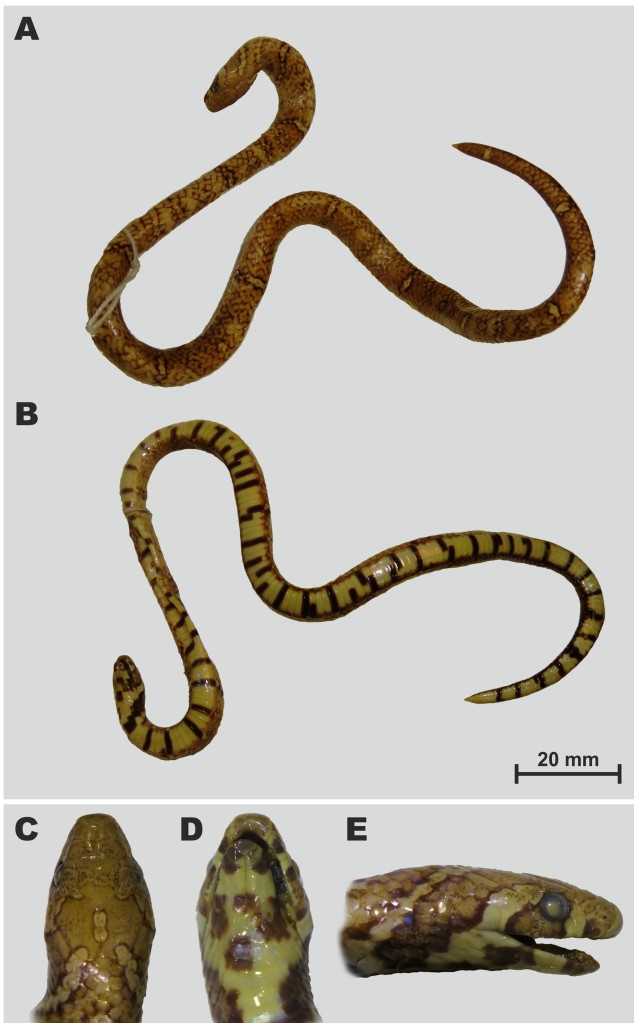

**Figure 8** **Holotype of *Oligodon annamensis* Leviton, 1953 from "Blao, Haut Donai, Station Agricole"**
**(now Bao Loc, Lam Dong Province, southern Vietnam) in preservative (USNM 90408, female).** (A)
General dorsal view; (B) general ventral view; (C) dorsal, (D) ventral, and (E) lateral head views. Photos by
Justin L. Lee; courtesy of National Museum of Natural History, Smithsonian Institution.

teeth; broad dark temporal streak; ground color on dorsum dark brown, 9–10+2–5 light
crossbars edged with black on dorsum and tail; vertebral stripe absent; and ventral surfaces
in life coral-red to orange with black transverse bars or quadrangular spots.

**Variation.** Morphological data of all presently known specimens of *O. annamensis* are
summarized in Table 4; color pattern of all *O. annamensis* specimens is remarkably similar
(Figs. 8–11). The holotype of *O. annamensis*, USNM 90408, corresponds well to the original
description by *Leviton (1953)* (Fig. 8), thus we do not provide its formal redescription.
The type specimen is a female with several morphological characters different from the
known male specimens (see Table 4): it has a relatively shorter tail, RTL 11.7% (vs. RTL
16.6–19.7% in three males), a greater number of ventrals, 170 (vs. 146–157 in males),
and a lesser number of subcaudals, 30 (vs. 43–46 in males). The second already known

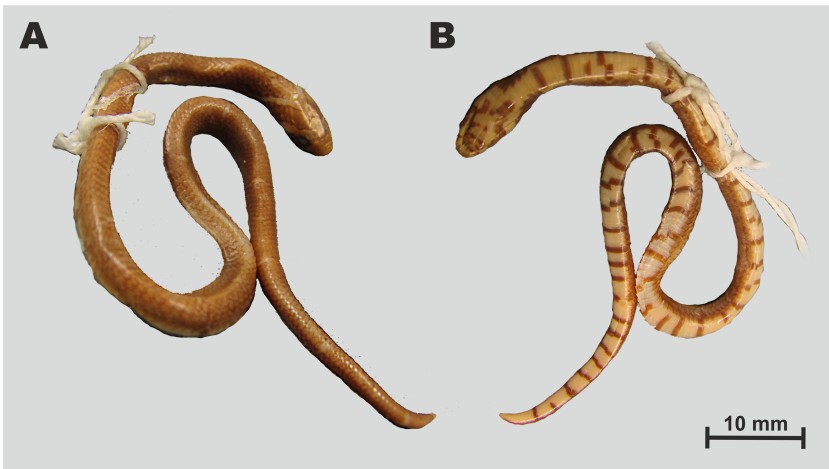

**Figure 9** Specimen of *Oligodon annamensis* Leviton, 1953 from "Blao, Haut Donai, Station Agricole" (now Bao Loc, Lam Dong Province, southern Vietnam) in preservative (MNHN 8815, male). (A) Dorsal view; (B) ventral view. Photos by Patrick David.

specimen of *O. annamensis*, MNHN 8815, a subadult male, was described in detail by *Leviton (1960)* (Fig. 9). Though in general morphology of MNHN 8815 corresponds well to the description by *Leviton (1960)*, we found several differences in scale counts: MNHN 8815 has 146 ventrals + 2 preventrals (vs. 159 ventrals, as stated by *Leviton (1960)* (courtesy of P. David). The reasons behind such significant differences in scale counts remain unclear; this result further underlines the importance of double-checking specimens preserved in historical collections in taxonomic practice.

The newly reported specimen of *O. annamensis* from Vietnam, ZMMU R-14304, was collected from Chu Yang Sin NP in Dak Lak Province at the northern edge of Langbian Plateau (see Fig. 1, locality 3). This specimen is an adult male and has the largest total length of all known *O. annamensis* specimens (412 mm); in scalation and coloration characters it agrees very well with the original description (*Leviton, 1953*) and the description of male specimen by *Leviton (1960)* (see Table 4). The tail of ZMMU R-14304 was dissected for examination of hemipenial structures; in full accordance with description by *Leviton (1960)* this specimen had deeply bilobed hemipenes each bearing two long and thin appendages seen in situ (papillae *sensu Smith, 1943*), reaching the 20th subcaudal. Coloration of ZMMU R-14304 in life is shown in Fig. 10; among other features, the characteristic coral-red background coloration of the ventral surfaces and black quadrangular spots forming complete transverse bars appear to be diagnostic from *Oligodon rostralis* **sp. nov.** (vs. in life ventral surfaces cream-white, black spots do not form transverse bars in the new species).

We present additional morphological information (see Table 4) and photos in life (Fig. 11) of the single known Cambodian specimen of *O. annamensis* CBC 01899 (see Fig. 1, locality 4) described by *Neang & Hun (2013)*. Based on relative tail length (16.6%) this specimen is identified as male. In accordance with earlier results of *Neang & Hun (2013)* it

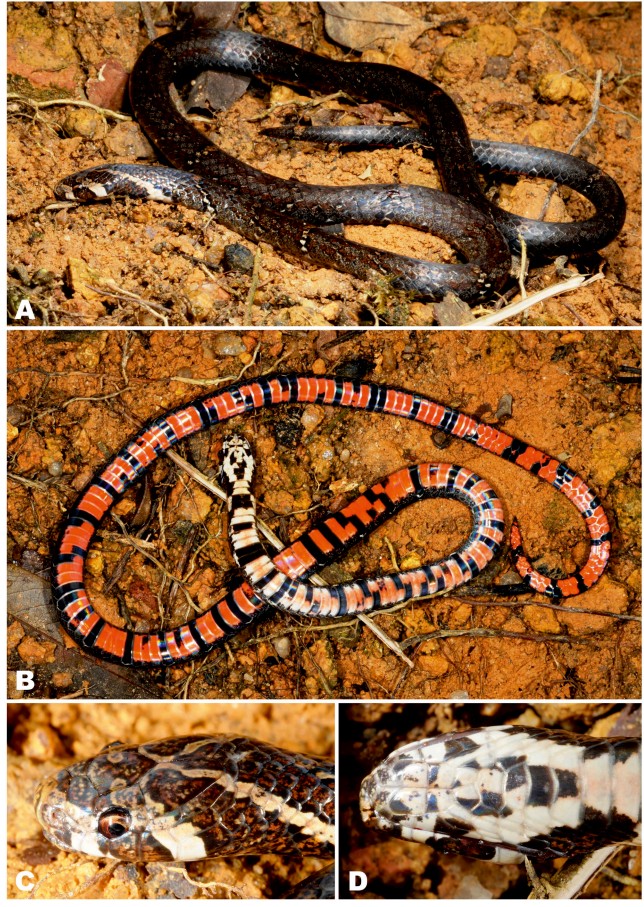

**Figure 10** **Specimen of *Oligodon annamensis* Leviton, 1953 from Chu Yang Sin NP, Dak Lak Province, southern Vietnam, in life (ZMMU R-14304, male).** (A) General dorso-lateral view; (B) general ventral view; (C) head in dorsal view; (D) head in ventral view. Photos by Nikolay A. Poyarkov.

shows certain morphological differences from the Vietnamese specimens, namely: having 6/5 supralabials of which 3–4/2–3 touching the orbit (vs. 6/6 and 3–4/3–4 in Vietnamese specimens); infralabials I–III contacting chin shields (vs. I–IV in Vietnamese specimens); posterior temporal single (vs. two posterior temporals in Vietnamese specimens); ventral coloration in life orange red with black markings not forming transverse bars, see Fig. 11B (vs. coral-red belly getting lighter anteriorly; black markings form numerous transverse bars in Vietnamese specimen, see Fig. 10B).

**Distribution.** To date *O. annamensis* is reliably known from two provinces of southern Vietnam (Lam Dong and Dak Lak), where it was recorded in montane forests of Langbian Plateau at elevations around 1,000 m a.s.l., and from similar elevations in montane forests of Phnom Samkos Mt. in the western part of the Cardamoms, Pursat, Cambodia. The record of *O. annamensis* from Dak Lak Province is a range extension and the first provincial record of this species.

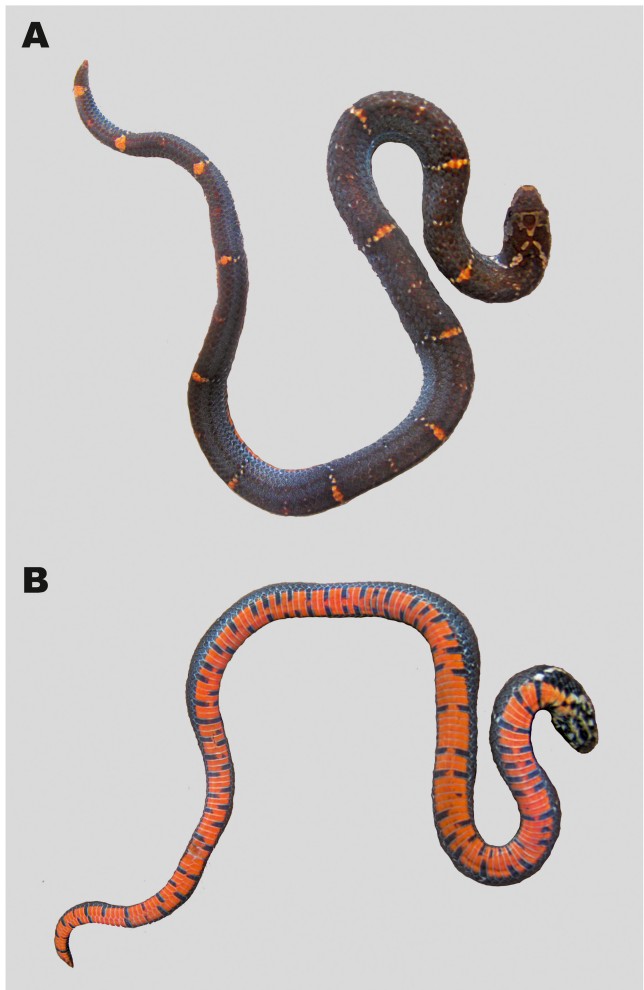

**Figure 11 Specimen of *Oligodon annamensis* Leviton, 1953 from Phnom Samkos WS, Dak Lak Pursat, Cardamom Mountains, Cambodia, in life (CBC 01899, male).** (A) Dorsal view; (B) ventral view. Photos by Hun Seiha.

**Etymology**. The specific name "*annamensis*" is a Latin adjective derived from "Annam", the historical name of Truong Son Mountains and central Vietnam. Common name in English: "*Leviton's kukri snake*" (English), "*Răn khiêm trung bo*" (Vietnamese), and "*Annamskiy oligodon*" (Russian).

## DISCUSSION

Our study reports on a new species of *Oligodon* from southern Vietnam, *Oligodon rostralis* **sp. nov.**, and provides new data on distribution, taxonomy and phylogenetic position of *O. annamensis*, including the first life photographs of this rare species and a range extension and first provincial record of *O. annamensis* for Dak Lak Province of Vietnam. We also confirm the previous identification of a specimen from Cardamom Mountains in Cambodia (*Neang & Hun, 2013*) as *O. annamensis* based on genetic and morphological

lines of evidence. Despite the observed minor morphological differences and geographic isolation, genetic differentiation between Cambodian and Vietnamese populations of *O. annamensis* is quite small and corresponds to common intraspecific levels of divergence in snakes ($p = 0.9\%$, see Table 3). Hence, *O. annamensis* has a disrupted range confined to Langbian Plateau in the east and to Cardamom Mountains in the west and separated by the Mekong River valley. Interestingly, a similar distribution pattern was recently reported for a number of lizard taxa inhabiting Indochina (e.g., *Grismer et al., 2019*; *Poyarkov et al., 2019a*), but was never recorded in Indochinese amphibians (*Geissler et al., 2015b*).

The genus *Oligodon* is traditionally classified in informal species groups on the basis of the hemipenial morphology, number of dorsal scale rows and other characters (*Smith, 1943*; *David, Das & Vogel, 2008*; *David et al., 2012*; *Vassilieva et al., 2013*; *Vassilieva, 2015*). The role of hemipenial morphology in delimiting clades within *Oligodon* was also partially confirmed based on phylogenetic analysis by *Green, Orlov & Murphy (2010)*. Among the species with available data on hemipenial morphology, only the species groups of *O. taeniatus* and *O. cyclurus* have bilobed hemipenes, while in other groups copulative organs are unilobed (*Green, Orlov & Murphy, 2010*). *Oligodon rostralis* **sp. nov.** shows a significant morphological similarity to *O. annamensis*—a species with unclear phylogenetic position. *Leviton (1960)*, describing hemipenial morphology of the only known male specimen, showed that *O. annamensis* has deeply bilobed hemipenis with papillae, basing on what he proposed that this species may be a part of the "*taeniatus-cyclurus-* complex" (*Leviton, 1953*; *Leviton, 1960*). Our observations on additional specimens of *O. annamensis* (see above) confirm the presence of deeply bifurcated hemipenes with papillae in this species. *Oligodon rostralis* **sp. nov.** also showed a forked hemipenis morphology, though lacking papillae. Our phylogenetic analysis suggests sister relationships between *Oligodon rostralis* **sp. nov.** and *O. annamensis* and places these two species in one clade with the members of the "*taeniatus-cyclurus-* complex", therefore confirming earlier hypothesis of *Leviton (1953)*; *Leviton (1960)*. Finally, our phylogeny also suggests that *O. lacroixi* is a sister species of *O. eberhardti* and is not closely related to *O. annamensis* or *Oligodon rostralis* **sp. nov.** despite certain morphological similarity between these species (*Orlov et al., 2010*).

The description of *Oligodon rostralis* **sp. nov.** brings the number of *Oligodon* species known for Vietnam to 24, thus making the country a local center of *Oligodon* diversity in Southeast Asia. Our work provides further evidence on high herpetofaunal diversity and endemism in Langbian Plateau, which mostly has been discovered only recently (e.g., *Chen et al., 2018*; *Duong et al., 2018*; *Geissler et al., 2015a*; *Geissler et al., 2015b*; *Hartmann et al., 2013*; *Nazarov et al., 2012*; *Orlov, Nguyen & Ho, 2008*; *Orlov et al., 2012*; *Pauwels et al., 2018*; *Poyarkov et al., 2014*; *Poyarkov et al., 2015a*; *Poyarkov et al., 2015b*; *Poyarkov et al., 2017*; *Poyarkov et al., 2018*; *Poyarkov et al., 2019a*; *Poyarkov et al., 2019b*; *Poyarkov & Vasilieva, 2011*; *Rowley et al., 2010*; *Rowley et al., 2011*; *Rowley et al., 2016*; *Stuart et al., 2011*; *Vassilieva et al., 2014*). Despite the impressive increase in species discoveries in the recent years, many isolated montane areas of the Truong Son Mountains, such as the Langbian Plateau, still remain insufficiently studied and likely cradle even more unknown biodiversity. The need for further biodiversity exploration in southern Indochina is urgent given the ongoing loss of natural habitats due to such intensifying threats as logging,

agricultural pressure, road construction and other anthropogenic activities (*De Koninck, 1999*; *Laurance, 2007*; *Meyfroidt & Lambin, 2008*; *Kuznetsov & Kuznetsova, 2011*). Further studies on herpetofaunal biodiversity in this region are immediately required for elaboration of effective conservation measures.

## CONCLUSIONS

Here, we present new molecular sequence data and an updated mtDNA genealogy for the genus *Oligodon*, one of the most species rich groups of Asian snakes. We confirm the presence of four main clades within the genus *Oligodon*, and for the first time report on the phylogenetic placement of several poorly known *Oligodon* species, including *O. annamensis* and *O. lacroixi*. We analyze all available collection material of *O. annamensis* from southern Vietnam and Cambodia and confirm the earlier assignation of Cambodian population from Cardamom Mountains to this species based on both morphological and molecular lines of evidence. Finally, we report on a new species of *Oligodon* from southern Vietnam, known from a single male specimen. *Oligodon rostralis* **sp. nov.** is distinct from all other congeners in a number of morphological diagnostic characters and is morphologically and phylogenetically most closely related to *O. annamensis*, from which it can be easily distinguished in scalation, coloration and mtDNA sequences. We analyze available morphological data on *Oligodon* species with 15 or 13 dorsal scale rows occurring in the mainland Asia, and discuss phylogenetic relationships among them. We provide further evidence for an unprecedented herpetofaunal diversity and endemism in Langbian Plateau, Southern Vietnam.

## ACKNOWLEDGEMENTS

The authors are grateful to Andrei N. Kuznetsov and Leonid P. Korzoun for support and organization of fieldwork. We want to thank Japan International Cooperation (JICA) for supporting the field work in Bidoup–Nui Ba National Park through the Sustainable Natural Resource Management Project (SNRM). Further, we thank the managers and staffs of Nippon Koei Co., Ltd. and Bidoup–Nui Ba National Park for their kind cooperation. We sincerely thank our Vietnamese colleagues Nguyen Dang Hoi, Hoang Minh Duc and Le Xuan Son for help and continued support. For permission to study specimens under her care we thank Valentina F. Orlova (ZMMU); Justin Lee (USNM) and Patrick David (MNHN) and Nikolai L. Orlov (ZISP) provided important information on several *Oligodon* specimens and useful criticism. H. N. Nguyen would like to give special thanks to his supervisor - Dr. Si-Min Lin (NTNU) for his enormous support with academic advice. We are grateful to Patrick David (MNHN), Seiha Hun (Phnompenh), Justin L. Lee (USNM), Jenna L. Welch (USNM) and the National Museum of Natural History, Smithsonian Institution, for providing us photographs of *Oligodon* specimens. We are thankful to Gernot Vogel, Justin L. Lee, Patrick David and an anonymous reviewer for their useful comments which allowed us to significantly improve the earlier versions of the manuscript.

### Funding

The study was completed with financial support from the Russian Science Foundation (RSF grant No 19-14-00050) to Nikolay A. Poyarkov (fieldwork, molecular phylogenetic analyses), and partial financial support from Ministry of Science and Technology, Taiwan (MOST 108-2311-B-003-001-MY3) to Hung Ngoc Nguyen (fieldwork, specimen examination). Dr. Si-Min Lin (NTNU) provided funding support to Hung Ngoc Nguyen. Japan International Cooperation (JICA) supported the field work in Bidoup–Nui Ba National Park through the Sustainable Natural Resource Management Project (SNRM). The funders had no role in study design, data collection and analysis, decision to publish, or preparation of the manuscript.

### Grant Disclosures

The following grant information was disclosed by the authors:
Russian Science Foundation: 19-14-00050.
Ministry of Science and Technology, Taiwan: MOST 108-2311-B-003-001-MY3.
Japan International Cooperation (JICA) supported the field work in Bidoup–Nui Ba National Park through the Sustainable Natural Resource Management Project (SNRM).

### Competing Interests

Nikolay A. Poyarkov is an Academic Editor for PeerJ.

### Author Contributions

- Hung Ngoc Nguyen and Platon V. Yushchenko analyzed the data, conceived and designed the experiments, performed the experiments, prepared figures and/or tables, authored or reviewed drafts of the paper, and approved the final draft.
- Bang Van Tran and Thy Neang analyzed the data, performed the experiments, prepared figures and/or tables, authored or reviewed drafts of the paper, and approved the final draft.
- Linh Hoang Nguyen performed the experiments, prepared figures and/or tables, authored or reviewed drafts of the paper, and approved the final draft.
- Nikolay A. Poyarkov analyzed the data, conceived and designed the experiments, prepared figures and/or tables, authored or reviewed drafts of the paper, and approved the final draft.

### Animal Ethics

The following information was supplied relating to ethical approvals (i.e., approving body and any reference numbers):

The Institutional Ethical Committee of Southern Institute of Ecology, Vietnamese Academy of Science and Technology (certificate number 114/QD-STHMN of November 8, 2016).

## Field Study Permissions

The following information was supplied relating to field study approvals (i.e., approving body and any reference numbers):

The fieldwork in Bidoup–Nui Ba NP was conducted under scope of the contract between Sustainable Nature Resource Management Project (SNRM) under Japan International Cooperation Agency and Southern Institute of Ecology to perform the "Biodiversity Baseline Survey" project of September 24, 2018. The fieldwork in Dak Lak Province was funded by the Joint Russian-Vietnamese Tropical and Technological Center (JRVTTC) and was conducted under permission of the Bureau of Forestry, Ministry of Agriculture and Rural Development of Vietnam (permits Nos. 170/ TCLN–BTTN of 07/02/2013; 400/TCLN-BTTN of 26/03/2014; 831/TCLN–BTTN of 05/07/2013) and of local administration, including the Forest Protection Department of the Peoples' Committee of Dak Lak Province (permit No. 388/SNgV-LS of 24/04/2019).

## DNA Deposition

The following information was supplied regarding the deposition of DNA sequences:

12S rRNA–16S rRNA mtDNA fragment and cytochrome b genes sequences are available at GenBank: MN395601–MN395604 and MN396762–MN396765.

## Data Availability

Specimens examined in this study are deposited in herpetological collections of the following museums:

1. The Department of Zoology, Southern Institute of Ecology (SIEZC), Ho Chi Minh City, Vietnam (SIEZC 20201, holotype of *Oligodon rostralis* sp. nov.);

2. Centre for Biodiversity Conservation of the Royal University of Phnom Penh (CBC RUPP), Phnom Penh, Cambodia (CBC 01899, *Oligodon annamensis*);

3. United States National Museum (USNM), Washington, D.C., USA (USNM 90408, holotype of *Oligodon annamensis Leviton, 1953*);

4. Museum National d'Histoire Naturelle (MNHN), Paris, France (MNHN 8815, *Oligodon annamensis*);

5. Zoological Museum of Lomonosov Moscow State University (ZMMU), Moscow, Russia (ZMMU R-14304, *Oligodon annamensis*).

## New Species Registration

The following information was supplied regarding the registration of a newly described species:

Publication LSID: urn:lsid:zoobank.org:pub:51B851C2-5D34-4065-86EA-CF18DDD94419

*Oligodon rostralis* LSID: urn:lsid:zoobank.org:act:37E1AAEA-1AF1-4A17-B2A1-D926ECAEB33F.

## Supplemental Information

Supplemental information for this article can be found online at http://dx.doi.org/10.7717/peerj.8332#supplemental-information.

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
