# Peer review of "A new species of Oligodon Fitzinger, 1826 from the Langbian Plateau, southern Vietnam, with additional information on Oligodon annamensis Leviton, 1953 (Squamata: Colubridae)"

_PeerJ, doi:10.7717/peerj.8332_

## Round 0.1 · original submission · Major Revisions

Dear authors

Your ms is of interest, but our reviewers request several changes. Please follow their recommendations, as your revision will be reviewed by the same reviewers.

Kind regards

Michael Wink
Academic Editor

Reviewer 1 ·

Basic reporting

no comment

Experimental design

no comment

Validity of the findings

no comment

Additional comments

This is well written manuscript based on extensive morphological and molecular comparisons and fieldwork to distinguish the new species in question. There is no doubt that this is a valid species and the authors have made a detailed description. There is some minor weakness in this paper, e.g using body ratio based on one sample to distinguish it from other congeners. This can be modified as per the suggestions I have made. There are other suggested minor changes which i have pointed out in the MS.

Annotated reviews are not available for download in order to protect the identity of reviewers who chose to remain anonymous.

·

Basic reporting

The paper is very interesting

Experimental design

no comment

Validity of the findings

no comment

Additional comments

I made my fe corrections with track back in the text, so I can sent the word file, I cannot upload it as DOCX.
The tables and figures are fine, with one mistake inside:
Table 5:
Oligodon eberharti
Change to Oligodon eberhardti

·

Basic reporting

The english in the manuscript requires revision, some phrasing and grammar needs to be corrected, key points are addressed in my comments. Literature references are fine, a few sources need to be checked for consistent formatting. Raw data is missing, mainly the raw tree used in the study. Photographs are in poor resolution or are mislabelled and need to be corrected.

Experimental design

Description of the new species and comparisons is straightforward. I do not like the way the account of new data on Oligodon annamensis is structured. I prefer that the authors re-write this section as a redescription. The specimen descriptions can be lumped together more and this will help organize the manuscript better. The species Oligodon lacroixi is not talked about at all in the paper despite its presentation a new DNA sequence. This needs to be addressed more or removed from the manuscript because it is not a close phylogenetic relative anyways.

Validity of the findings

The new species is based off a single male specimen, but given its phylogenetic position based on multiple loci and the morphological differences, I do not have any objections to its validity or diagnosis.

Additional comments

The article "A new species of Oligodon Fitzinger, 1826 from Langbian Plateau, southern Vietnam, with additional information on Oligodon annamensis Leviton, 1953 (Squamata: Colubridae)" needs important revisions before being accepted into this journal. The following issues need to be resolved in the next version of the manuscript.

1. Reformat the account on Oligodon annamensis as a redescription or a seperate species account. Discussion section for this species looks unorganized and would be streamlined if redone as a formal description of variation.
2. Remove or add more information on Oligodon lacroixi since it is a novel sequence you are reporting in the study.
3. Consistent use and clarification of hemipenial terminology used (for example: papillae as an in-situ appendage vs. papillae as everted hemipenis ornamentation; "forked and "unforked" vs. bilobed and unilobed; use of bifurcation to describe hemipenis shape vs. use of bifurcation to describe the sulcus spermaticus). Also the pictures of the hemipenis need to be redone to show ornamentation/structure better.

Additional comments and suggestions are outlined in the attached PDF and should be addressed.

– Justin L. Lee

---

## Round 0.2 · accepted · Accept

Dear authors,

Congratulations. Your revision is adequate and therefore, we can accept your ms.

Kind regards,
Michael Wink
Academic Editor

·

Basic reporting

See previous review

Experimental design

See previous review

Validity of the findings

See previous review

Additional comments

The authors of this paper have drastically improved their manuscript according to comments made by the reviewers and myself. The revisions made are adequate, the paper reads well and mistakes in earlier drafts were addressed.

There are a few instances in the tables, the species diagnosis for Oligodon rostralis and in the discussion where the hemipenial term "forked" is used when it should be "bilobed", and in line 774 in the discussion, "unilobed" is used when it should be "unforked". This may need to be revised for consistency. Other than that, I believe the manuscript is ready for acceptance pending comments from other reviewers.

– Justin Lee